# Endoplasmic reticulum stress enhances fibrosis through IRE1α-mediated degradation of miR-150 and XBP-1 splicing

Femke Heindryckx[1,†], François Binet[1,†], Markella Ponticos[2], Krista Rombouts[3], Joey Lau[1], Johan Kreuger[1,‡] & Pär Gerwins[1,4,*,‡]

## Abstract

ER stress results in activation of the unfolded protein response and has been implicated in the development of fibrotic diseases. In this study, we show that inhibition of the ER stress-induced IRE1α signaling pathway, using the inhibitor 4μ8C, blocks TGFβ-induced activation of myofibroblasts *in vitro,* reduces liver and skin fibrosis *in vivo,* and reverts the fibrotic phenotype of activated myofibroblasts isolated from patients with systemic sclerosis. By using IRE1α[−/−] fibroblasts and expression of IRE1α-mutant proteins lacking endoribonuclease activity, we confirmed that IRE1α plays an important role during myofibroblast activation. IRE1α was shown to cleave miR-150 and thereby to release the suppressive effect that miR-150 exerted on αSMA expression through c-Myb. Inhibition of IRE1α was also demonstrated to block ER expansion through an XBP-1-dependent pathway. Taken together, our results suggest that ER stress could be an important and conserved mechanism in the pathogenesis of fibrosis and that components of the ER stress pathway may be therapeutically relevant for treating patients with fibrotic diseases.

**Keywords** endoplasmic reticulum stress; fibrosis; liver cirrhosis; myofibroblast; scleroderma

**Subject Categories** Digestive System; Pharmacology & Drug Discovery; Skin

## Introduction

Fibrosis is characterized by an excessive accumulation of fibrous connective tissue that typically occurs in response to chronic inflammation caused by toxic substances, infections, mechanical injury, or as a result of autoimmune reactions (Darby & Hewitson,

2007; Kramann *et al*, 2013; Kendall & Feghali-Bostwick, 2014). Progressive accumulation of connective tissue ultimately leads to disruption of tissue architecture and organ dysfunction that can be fatal as seen in end-stage liver disease, kidney disease, pulmonary fibrosis, and in heart failure. Chronic autoimmune diseases such as scleroderma, ulcerative colitis, Crohn's disease, and rheumatoid arthritis are additional examples of diseases where fibrosis is a prominent pathological feature (Gilbane *et al*, 2013; Li & Kuemmerle, 2014; Pattanaik *et al*, 2015). Despite that fibrosis can affect nearly every tissue in the body and is a major cause of morbidity and mortality in a variety of diseases, there are few treatments that specifically target the pathogenesis of fibrosis.

Although the etiology and causative mechanisms may vary widely, fibrosis always involves excessive production and deposition of collagen and other extracellular matrix proteins by activated myofibroblasts. Myofibroblasts express alpha-smooth muscle actin (αSMA) and are formed and activated as part of a repair mechanism triggered by tissue damage (Kendall & Feghali-Bostwick, 2014). Several cell types have been shown to serve as progenitors for myofibroblasts: residential fibroblasts, epithelial cells (through epithelial-to-mesenchymal transition) as well as blood-borne mesenchymal progenitor cells called fibrocytes. In the liver, activated myofibroblasts derived from hepatic stellate cells (HSCs) are the principal fibroblastic cell type responsible for matrix deposition.

The formation and function of myofibroblasts is promoted by a number of secreted soluble factors such as cytokines, growth factors, and matrix-related factors that act in synergy in the development of fibrosis (Kramann *et al*, 2013). TGFβ is considered a key factor that coordinates the interplay between parenchymal cells, inflammatory cells, and myofibroblasts (Leask & Abraham, 2004). Accordingly, overexpression of TGFβ in transgenic mice results in multi-organ fibrosis (Sime *et al*, 1997; Kanzler *et al*, 1999; Rahmutula *et al*, 2013).

Homeostatic regulation of protein folding in the endoplasmic reticulum (ER) is under the control of three evolutionary conserved

1 Department of Medical Cell Biology, Uppsala University, Uppsala, Sweden
2 Centre for Rheumatology and Connective Tissue Diseases, University College London, London, UK
3 Institute for Liver and Digestive Health, University College London, London, UK
4 Department of Radiology, Uppsala University Hospital, Uppsala, Sweden
*Corresponding author. Tel: +46 184714079; Fax: +46 184714059; E-mail: par.gerwins@mcb.uu.se
†These authors contributed equally to this work as first authors
‡These authors contributed equally to this work as last authors

pathways: IRE1α-XBP-1, PERK-eIF2α, and ATF6 (Schroder & Kaufman, 2005). These pathways, also referred to as the unfolded protein response (UPR), are activated in response to ER stress, which occurs when cells have to cope with an increased burden in the protein folding machinery. Importantly, the UPR makes life/death decisions for the cell and the final outcome of ER stress is either recovery and survival or apoptosis depending on the severity and duration of ER stress (Sovolyova *et al*, 2014). All three pathways of the UPR form a coordinated reaction to the accumulation of unfolded proteins and several studies have demonstrated that there is cross talk between the different pathways (Yamamoto *et al*, 2004; Arai *et al*, 2006; Adachi *et al*, 2008). ER stress and activation of the UPR can be caused by mechanical injury, inflammation, genetic mutations, infections, and oxidative stress, and has been implicated in diseases that result in fibrotic remodeling of internal organs, such as chronic liver diseases (Galligan *et al*, 2012; Shin *et al*, 2013; Ji, 2014), pulmonary fibrosis (Baek *et al*, 2012; Tanjore *et al*, 2012, 2013), kidney fibrosis (Chiang *et al*, 2011), cardiovascular disease (Spitler & Webb, 2014), and inflammatory bowel disease (Bogaert *et al*, 2011; Cao *et al*, 2013).

IRE1α is a kinase located in the ER membrane that transmits information across the ER lipid bilayer (Tirasophon *et al*, 2000). Increased ER protein load and presence of unfolded proteins leads to the dissociation of the ER chaperone GRP78/BiP from IRE1α. The free IRE1α then dimerize and autophosphorylate, which leads to activation of the IRE1α ribonuclease domain in the cytosol. Regulated IRE1α-dependent decay of mRNA (RIDD) reduces both protein translation and import of proteins into the ER (Hollien & Weissman, 2006). In addition, IRE1α has the ability to splice XBP-1 mRNA, which is required for the translation of active XBP-1. After transport into the nucleus, XBP-1 binds to UPR promoter elements to initiate expression of genes that enhance the ability of the ER to cope with unfolded proteins, including molecular chaperones like BiP and transcription factors such as CHOP. ER stress is in addition associated with enlargement of the ER volume, which has been interpreted as an adaptive mechanism to increase protein folding capacity (Sriburi *et al*, 2004).

Besides degrading mRNA (Binet *et al*, 2013), it was recently shown that IRE1α also has the ability to degrade microRNAs (miRs) (Upton *et al*, 2012). miRs are short noncoding RNA oligonucleotides consisting of 17–25 nucleotides that generally act to inhibit gene expression by binding to complementary sequences in the 3′-untranslated region of target mRNAs, either to repress mRNA translation or to induce mRNA cleavage. A number of cellular functions such as proliferation, differentiation, and apoptosis are regulated by miRs, and aberrant miR expression is observed in a variety of human diseases including fibrosis (Bowen *et al*, 2013).

Inhibitors that specifically target individual components of the UPR have recently been developed. The selective inhibitor 4μ8C that stably binds to lysine 907 in the IRE1α endoribonuclease domain has been shown to inhibit both RIDD activity and XBP-1 splicing (Cross *et al*, 2012). High levels of 4μ8C cause no measurable toxicity in cells and concentrations ranging from 80 to 128 μM of 4μ8C completely block XBP-1 splicing without affecting IRE1α kinase activity (Cross *et al*, 2012). The inhibitor 4μ8C thus represents both a potential drug and an important tool to delineate the functions of IRE1α *in vivo* as IRE1α-knockout mice die during embryonic development.

In this report, we show that inhibition of IRE1α prevents activation of myofibroblasts and reduces fibrosis in animal models of liver and skin fibrosis. Our finding that pharmacological inhibition of IRE1α could revert the profibrotic phenotype of activated myofibroblasts isolated from patients with scleroderma indicates that ER stress inhibitors should be taken into consideration when developing new strategies for the treatment of fibrotic diseases.

## Results

### The RNase activity of IRE1α is required for activation of myofibroblasts

Enhanced secretion of extracellular matrix proteins as well as increased αSMA expression are typical features of fibroblast to myofibroblast conversion (Baum & Duffy, 2011). Treatment of human fetal lung fibroblasts (HFL1) with TGFβ resulted in formation of myofibroblasts as judged by the increased expression of αSMA (Fig 1A) and collagen I (Fig 1B). Since increased collagen I production may increase the protein folding demand and activate the UPR, we measured the expression of several ER stress pathway components and found increased mRNA expression of CHOP and BiP (Fig 1C) as well as increased protein levels of CHOP (Fig 1D) and increased splicing of XBP-1 (Fig 1C and E). To investigate whether this activation of ER stress, and specifically the activation of IRE1α, is involved in myofibroblast activation, we inhibited IRE1α function using the inhibitor 4μ8C, which blocked TGFβ-induced activation of fibroblasts as measured by expression of mRNA and protein levels of αSMA (Fig 1A and F) and collagen I (Fig 1B and G). Further, TGFβ-induced expression of αSMA mRNA was reduced in cells overexpressing two different mutant forms of IRE1α that lack RNase activity (Fig 1H) or when complete loss-of-function experiments were performed using IRE1α$^{-/-}$ MEF cells (Fig 1I).

### miR-150 exhibits anti-fibrotic effects and is directly cleaved by IRE1α

IRE1α has recently been shown to directly cleave miRs (Upton *et al*, 2012) and it was therefore hypothesized that 4μ8C could modulate fibrosis by preventing cleavage of miRs involved in activation of myofibroblasts. Decreased levels of miR-29 (Khalil *et al*, 2015), miR-148 (Van Keuren-Jensen *et al*, 2015), and miR-150 (Honda *et al*, 2013; Zheng *et al*, 2013) have been implicated in fibrotic tissue remodeling and were therefore selected for analysis. The combined treatment of HFL1 fibroblasts with TGFβ and 4μ8C led to a significant increase in miR-150 (Fig 2A), while the levels of miR-29 and miR-148 were unaffected (Fig EV1). These results suggested that an IRE1α-dependent signaling pathway could be responsible for regulating the levels of miR-150. This was further supported by the finding that overexpression of RNase inactive IRE1α mutants in HFL1 fibroblasts resulted in increased levels of miR-150 in response to TGFβ (Fig 2B). Further, miR-150 levels were also higher in TGFβ-treated IRE1α$^{-/-}$ MEF cells, as compared to WT cells (Fig 2C). To further investigate whether IRE1α-mediated cleavage of miR-150 contributes to TGFβ-induced myofibroblast activation, IRE1α$^{-/-}$

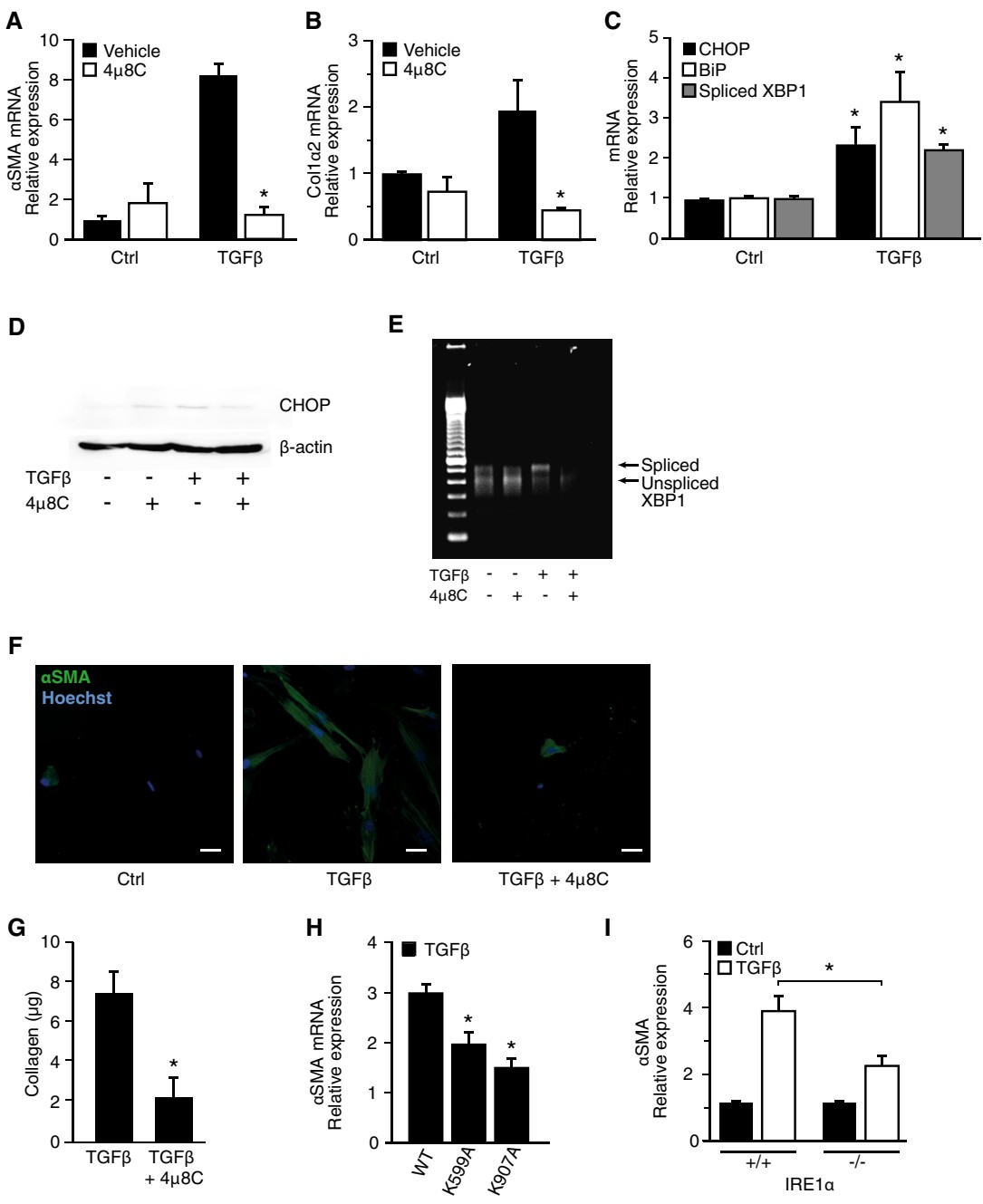

**Figure 1. The RNase activity of IRE1α is required for activation of myofibroblasts.**

A   αSMA mRNA levels in TGFβ-treated HFL1 fibroblasts and the effect of the IRE1α inhibitor 4μ8C. *P = 0.001.

B   Collagen 1α2 mRNA levels in TGFβ-treated HFL1 fibroblasts ± the IRE1α inhibitor 4μ8C. *P = 0.01.

C   mRNA levels of ER stress-related genes in TGFβ-treated HFL1 fibroblasts. *P CHOP = 0.045, *P Bip = 0.023, *P spliced XBP-1 P = 0.001, compared to corresponding control.

D   Western blot of CHOP protein levels in TGFβ-treated HFL1 cells ± the IRE1α inhibitor 4μ8C.

E   Agarose gel showing the presence of spliced and unspliced XBP-1 mRNA in TGFβ-treated HFL1 fibroblasts ± the IRE1α inhibitor 4μ8C.

F   Immunohistochemical staining of αSMA in TGFβ-treated HFL1 fibroblasts ± the IRE1α inhibitor 4μ8C. Scale bars = 20 μm.

G   Quantification of secreted soluble collagen proteins produced by TGFβ-treated HFL1 cells ± the IRE1α inhibitor 4μ8C. *P = 0.049.

H   αSMA mRNA expression in TGFβ-treated HFL1 cells overexpressing two RNase-dead mutants (K599A and K907A) of IRE1α. *P K599A = 0.013, *P K907A = 0.0008 compared to WT.

I   αSMA mRNA expression in TGFβ-treated IRE1α$^{-/-}$ or WT MEF cells. *P = 0.025.

Data information: Statistical significance was evaluated using Student's *t*-test. Significant differences are indicated with * and exact *P*-values are given. Error bars indicate s.e.m. *n* = 3 in (A–C) and (G–I).
Source data are available online for this figure.

MEF cells were transfected with an anti-miR-150-5p miRNA inhibitor, which was shown to suppress the levels of miR-150 (Fig 2D) and to increase αSMA mRNA in response to TGFβ (Fig 2E). Based on this, it was hypothesized that miR-150 is directly involved in the negative regulation of αSMA expression.

The transcription factor c-Myb represents the top target gene of miR-150 with 3 conserved miR-150 binding sites in the human mRNA according to TargetScan (Lewis *et al*, 2003). Several studies have shown that miR-150 suppresses c-Myb expression (Lin *et al*, 2008; Feng *et al*, 2014; Yang *et al*, 2015) and c-Myb in turn is known to increase αSMA expression (Kitada *et al*, 1997; Buck *et al*, 2000; Zheng *et al*, 2013). We therefore investigated whether miR-150 could regulate c-Myb and αSMA expression in HFL1 fibroblasts. Transfection of HFL-1 fibroblasts with a miR-150 expression plasmid (Fig 2F) resulted in reduced c-Myb and αSMA protein expression (Fig 2G and H) supporting an anti-fibrotic function of miR-150. To further establish the link between c-Myb and αSMA, HFL1 fibroblasts were transfected with siRNA targeting c-Myb (Fig 2I). In line with previous studies (Kitada *et al*, 1997; Buck *et al*, 2000; Zheng *et al*, 2013), silencing of c-Myb reduced TGFβ-induced αSMA expression (Fig 2J).

Sequence analysis of pre-miR-150 showed the presence of several putative IRE1α cleavage sites, including one positioned 10 bp from the 5′-terminus (Fig 2K). The possibility that IRE1α may directly cleave miR-150 was investigated using an *in vitro* cleavage assay, with $^{32}$P-labeled pre-miR-150 as a substrate and a synthetic stem-loop RNA corresponding to the XBP-1 substrate as a positive control. It was found that recombinant IRE1α cleaved pre-miR-150, which generated a small cleavage product of approximately 10 bp (Fig 2L). Cleavage of pre-miR-150 and the XBP-1 stem-loop control was almost completely inhibited by addition of 4μ8C, which confirmed that 4μ8C is an inhibitor of IRE1α RIDD activity (Fig 2L). Next, HFL1 cells were pre-treated with TGFβ and 4μ8C, and transcription inhibited using actinomycin D for 4 and 24 h to see whether the stability of miR-150 was affected. Indeed, 4μ8C was shown to slow down miR-150 degradation in TGFβ-treated

HFL1-cells (Fig 2M). Thus, degradation of miR-150 can be directly mediated by IRE1α, and loss of IRE1α RNase activity results in increased levels of miR-150, which results in c-Myb suppression and reduced TGFβ-induced αSMA gene expression.

### ER expansion in myofibroblasts is coupled to XBP-1 splicing

Activation of UPR pathways, in particular splicing of XBP-1, has been linked to ER expansion and increased secretory capacity (Sriburi *et al*, 2004; Bommiasamy *et al*, 2009). Therefore, we hypothesized that ER stress, through splicing of XBP-1 by IRE1α, mediates ER expansion and is at least partially required for TGFβ-induced activation of myofibroblasts and secretion of extracellular matrix. TGFβ-stimulation of HFL1 fibroblasts resulted in an enlargement of the ER (Fig 3A and B) in agreement with previous studies on myofibroblasts (Baum & Duffy, 2011). Treatment of cells with 4μ8C inhibited TGFβ-induced ER expansion suggesting a central role of IRE1α (Fig 3A and B). To specifically investigate the role of XBP-1, we treated HFL1 fibroblasts with siRNA targeting XBP-1, which caused an 85% reduction in XBP-1 gene expression (Fig 3C) and inhibited TGFβ-induced ER expansion (Fig 3A and B), as well as collagen secretion (Fig 3D). These observations suggest that the IRE1α-XBP-1 axis contributes to myofibroblast activation by expanding the ER in order to cope with increased protein folding demands.

### Pharmacological inhibition of IRE1α attenuates liver fibrosis

The UPR is activated in several liver diseases (Malhi & Kaufman, 2011). To investigate whether inhibition of IRE1α could reduce development of liver cirrhosis, we used a mouse model in which hepatic fibrosis was induced by alcohol and $CCl_4$ (Geerts *et al*, 2008). Mice developed typical bridging liver fibrosis with an increased METAVIR score reflecting F3-F4 cirrhosis (Bedossa & Poynard, 1996) (Fig 4A and B). Treatment with 4μ8C reduced fibrosis (Fig 4A and B) and the number of αSMA-positive

▶

---

**Figure 2. miR-150 exhibits anti-fibrotic effects and is directly cleaved by IRE1α.**

A   miR-150 levels in TGFβ-treated HFL1 fibroblasts and the effect of the IRE1α inhibitor 4μ8C. *$P$ = 0.046 compared to control.

B   miR-150 levels in HFL1 fibroblasts overexpressing two RNase-dead mutants (K599A and K907A) of IRE1α. *$P$ = 0.0043 (K599A), $P$ = 0.043 (K907A), compared to control.

C   miR-150 levels in TGFβ-treated WT (IRE1α $^{+/+}$) and IRE1α $^{-/-}$ MEF cells. *$P$ = 0.018. Results are expressed as fold change compared to the Ctrl.

D   miR-150 levels in IRE1α $^{-/-}$ MEF cells that were either untransfected (UT) or transfected with miRNA inhibitor control (siCtrl) or with a miR-150-5p miRNA inhibitor. *$P$ = 0.034 compared to UT.

E   αSMA mRNA expression in TGFβ-treated IRE1α $^{-/-}$ MEF cells that were either untransfected (UT) or transfected with miRNA inhibitor control (siCtrl) or with a miR-150-5p miRNA inhibitor. *$P$ = 0.043 compared to UT.

F   miR-150 levels in HFL1 fibroblasts transfected with a control plasmid (Ctrl) or a miR-150 expression plasmid (miR-150). *$P$ = 0.038.

G   Western blot of c-Myb in TGFβ-treated HFL1 fibroblasts that were either untransfected (UT), transfected with a control plasmid (pCtrl) or a miR-150 expression plasmid (miR-150). *$P$ = 0.0206 in UT Ctrl versus UT TGFβ and $P$ = 0.0126 in pCtrl Ctrl versus pCtrl TGFβ.

H   Western blot of αSMA in the same experimental setup as in (G). *$P$ = 0.0405 in UT Ctrl versus UT TGFβ, and $P$ = 0.0284 in pCtrl Ctrl versus pCtrl TGFβ.

I   c-Myb mRNA levels in untransfected HFL1 cells (UT) and in cells transfected with a control siRNA (siCtrl) or an siRNA targeting c-Myb (sicMyb). *$P$ = 0.00001 compared to UT.

J   αSMA mRNA expression in untransfected HFL1 cells (UT) and in cells transfected with a control siRNA (siCtrl) or with a siRNA targeting c-Myb (sicMyb). *$P$ = 0.0042 in UT Ctrl versus UT TGFβ.

K   Putative IRE1α cleavage sites in miR-150 based on sequence analysis are indicated with arrows.

L   *In vitro* cleavage assay with recombinant IRE1α and miR-150 (left) or XBP-1 (right).

M   miR-150 expression in HFL-1 cells that were pre-treated with TGFβ in the absence of presence of 4μ8C and then treated with actinomycin D (ActD) for 4 and 24 h. *$P$ = 0.027 compared to TGFβ.

Data information: Statistical significance was evaluated using Student's *t*-test. Significant differences are indicated with * and exact *P*-values given. Error bars indicate s.e.m. $n$ = 3 in (A–E, G–J, M) and $n$ = 4 in (F).
Source data are available online for this figure.

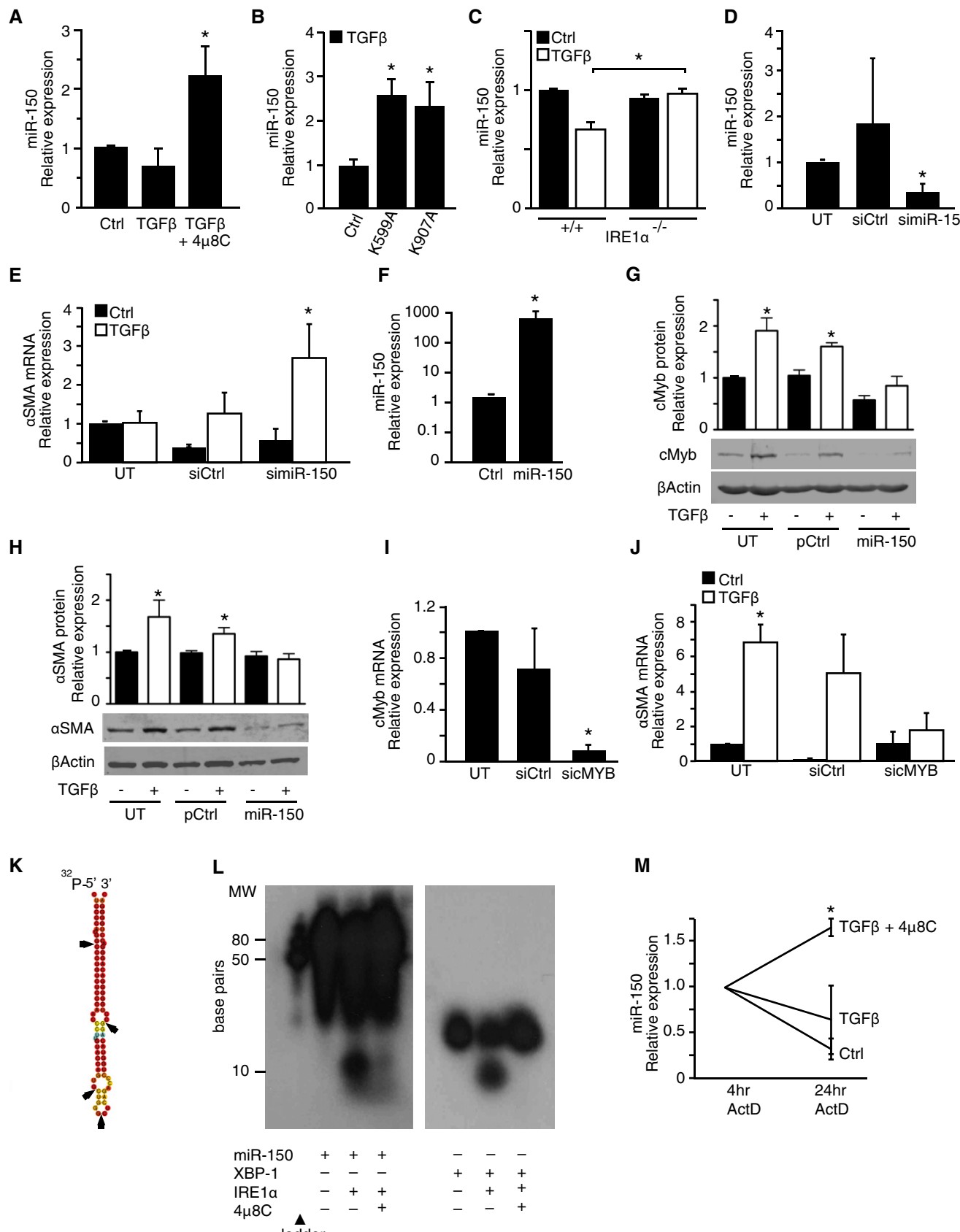

**Figure 2.**

myofibroblasts in pericentral areas (Fig 4C). Fibrotic livers typically have an increased size and weight and treatment with 4µ8C decreased the liver-to-body weight ratio to further support the

hypothesis that inhibition of the UPR reduces disease progression (Fig 4D). To verify that 4µ8C affected activation of the UPR, markers of the UPR pathway were measured. The mRNA levels of BiP

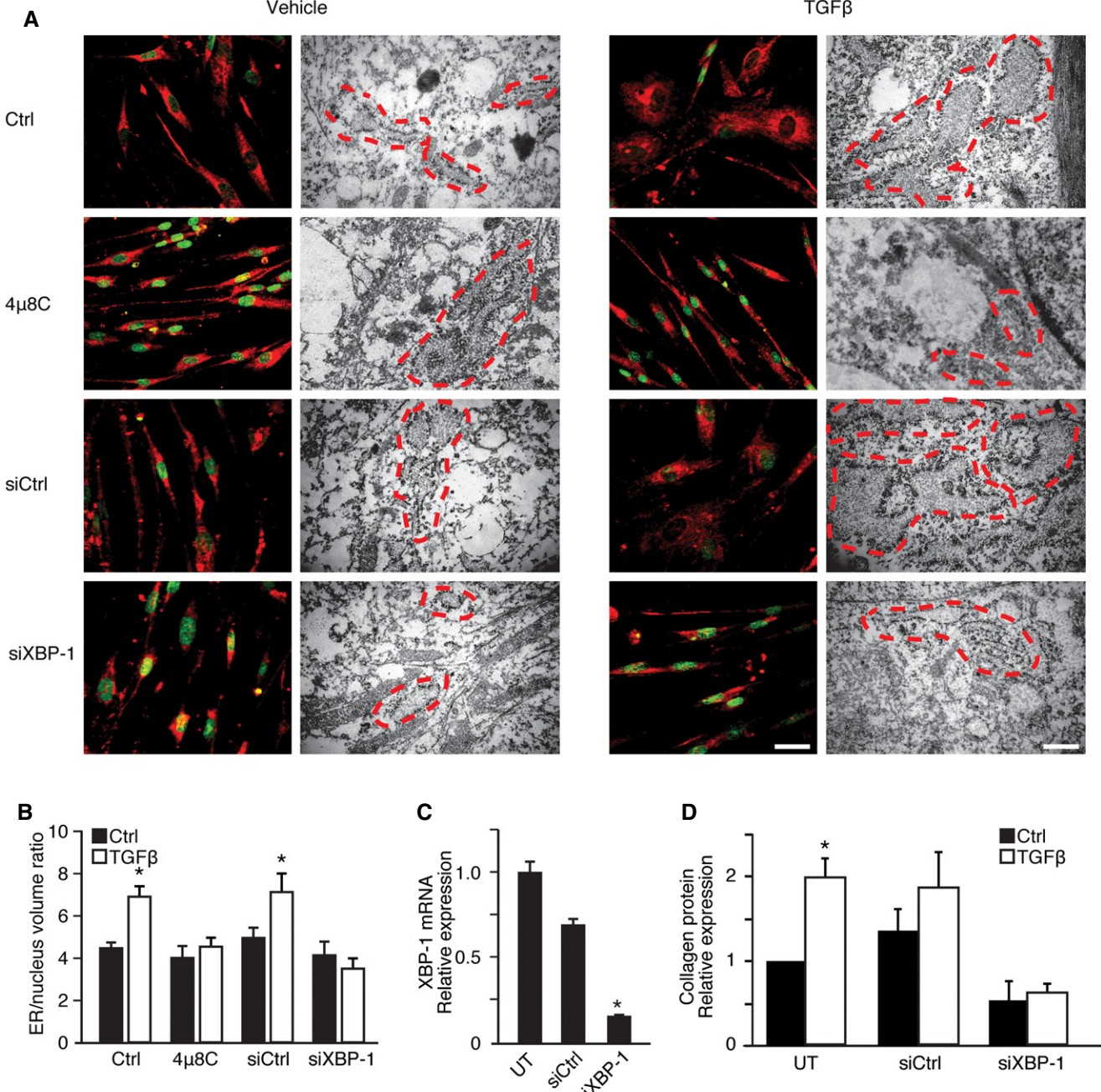

**Figure 3. ER expansion in myofibroblasts is coupled to XBP-1 splicing.**

A   Analysis of ER size in HFL1 fibroblasts treated with TGFβ in the absence (Ctrl) or presence of the IRE1α inhibitor 4µ8C, or in HFL1 cells transfected with a siRNA control siRNA (siCtrl) or with siRNA targeting XBP-1 (siXBP-1). The ER was visualized with ER-Tracker Red and nuclei stained with Hoechst (left, scale bars = 20 µm) or by electron microscopy (right, scale bars = 200 nm) where red lines mark the ER.

B   Quantification of ER size relative to the nucleus from the experimental setup in (A). *P = 0.0001 in Ctrl, and *P = 0.0223 in siCtrl.

C   XBP-1 mRNA expression in untransfected HFL1 cells (UT) and in cells transfected with control siRNA (siCtrl) or siRNA targeting XBP-1 (siXBP-1). *P = 0.0001.

D   Collagen protein levels in TGFβ-treated HFL1 fibroblasts transfected as described in (C). *P = 0.005.

Data information: Statistical significance was evaluated using Student's *t*-test. Significant differences are indicated with * and exact *P*-values given. Error bars indicate s.e.m. *n* = 3 in (B, C) and *n* = 4 in (D).

(Fig 4E) and splicing of XBP-1 (Fig 4F) were reduced in livers from mice treated with 4μ8C. In support of a model where miR-150 exerts anti-fibrotic effects, we found that miR-150 levels were increased in liver tissue isolated from 4μ8C-treated mice (Fig 4G).

Immunohistochemical staining of liver sections showed co-localization of the myofibroblast marker αSMA with the UPR markers BiP and CHOP, which suggests that activated HSCs were contributing to the observed increased activation of the UPR (Fig EV2A and B). In addition, HSCs were isolated from mice and *in vitro* treatment with 4μ8C inhibited TGFβ-induced expression of αSMA and BiP (Fig 4H–K) indicating that the fibrosis-reducing effect of 4μ8C is by inhibition of ER stress and reduced activation of HSCs. Similarly, 4μ8C inhibited TGFβ-induced collagen I and connective tissue growth factor (CTGF) mRNA expression in HSCs isolated from human liver (Fig 4L). These results suggest that inhibition of IRE1α RNase activity can attenuate liver fibrosis by hindering HSCs to become activated myofibroblasts.

### Inhibition of IRE1α reduces skin fibrosis

In order to investigate whether interference with ER stress at the level of IRE1α affects fibrosis in other models, and hence could be of more general applicability, we used a mouse model of skin fibrosis where TGFβ is continuously infused through subcutaneously implanted mini-osmotic pumps. TGFβ caused a fibrotic response in the hypodermis with an increased number of activated αSMA-expressing myofibroblasts, together with an increased amount of deposited collagen, which was significantly decreased in 4μ8C-treated mice (Fig 5A–C). Selective analysis of the hypodermis by laser capture microdissection (Fig 5D) confirmed that treatment with 4μ8C reduced αSMA mRNA expression (Fig 5E) and a tendency toward increased miR-150 levels (Fig 5F). Staining of skin sections from TGFβ-treated mice showed co-localization of the myofibroblast marker αSMA with the UPR markers BiP and CHOP in the hypodermis, which confirms activation of the UPR in myofibroblasts (Fig EV3A and B). Treatment of primary mouse skin fibroblasts with 4μ8C similarly inhibited TGFβ-induced αSMA expression (Fig 5G and H) and 4μ8C also decreased activation of the UPR (Fig 5I–K) and increased the level of miR-150 (Fig 5L). This

further supports that 4μ8C inhibits fibrosis by preventing activation of myofibroblasts.

### The fibrotic phenotype of myofibroblasts isolated from patients with scleroderma can be reverted by inhibition of IRE1α

Scleroderma, or progressive systemic sclerosis (SSc), is a connective tissue disease characterized by severe multi-organ fibrosis, which results from the excessive synthesis of extracellular matrix by myofibroblasts (Gilbane *et al*, 2013). To explore the possibility that IRE1α could be a potential therapeutic target for treatment of SSc, fibroblasts isolated from SSc patients were treated with 4μ8C. These experiments showed that pharmacological inhibition of IRE1α significantly decreased mRNA levels of CTGF (Fig 6A) and collagen I (Fig 6B) and soluble collagen protein expression (Fig 6C) in fibroblasts from both skin (dSSc) and lung (LSSc). Fibroblasts from SSc patients also showed increased levels of BiP, which were reduced after treatment with 4μ8C (Fig 6D). To further study the general applicability of treatment with 4μ8C, primary lung fibroblasts from patients with inflammatory lung diseases, such as cystic fibrosis (CF), chronic obstructive pulmonary disease (COPD), and asthma, were exposed to 4μ8C. CF, COPD, and asthma are diseases that cause chronic inflammation of the airways, involving myofibroblast activation and ECM deposition. Primary fibroblasts from asthma patients, but not fibroblasts isolated from CF or COPD patients, expressed significantly increased levels of αSMA and BiP compared to controls, which were decreased by treatment with 4μ8C (Fig 6E and F). These results suggest that inhibition of IRE1α not only can prevent formation and activation of myofibroblasts, but also can revert the fibrotic phenotype of myofibroblasts in cells isolated from patients with fibrotic disease.

## Discussion

There is increasing evidence that ER stress and activation of the UPR is a feature of fibrosis in different organs caused by different etiologies. On a cellular level, the myofibroblast is considered to have a central role in development of fibrosis, but little is known

---

**Figure 4.  Pharmacological inhibition of IRE1α attenuates liver fibrosis.**

A   The effect of the IRE1α inhibitor 4μ8C on CCl₄-induced liver fibrosis in C57BL/6 mice as measured by the histological METAVIR score. Ctrl ($n = 8$); Ctrl + 4μ8C ($n = 6$); CCl₄ ($n = 6$); CCl₄ + 4μ8C ($n = 8$). *$P = 0.0077$.

B   Representative liver tissue sections stained with sirius red for collagen (top) and αSMA (bottom). Scale bars = 50 μm.

C   Percentage of αSMA-positive cells in pericentral areas in livers from mice in the experimental setup in (A). *$P = 0.036$.

D   Liver-to-body weight ratio from the experimental setup in (A). *$P = 0.0031$. Ctrl ($n = 8$); Ctrl + 4μ8C ($n = 6$); CCl₄ ($n = 6$); CCl₄ + 4μ8C ($n = 8$).

E   BiP mRNA expression in liver tissue from mice presented in (A). *$P = 0.020$. Ctrl ($n = 5$); Ctrl + 4μ8C ($n = 5$); CCl₄ ($n = 5$); CCl₄ + 4μ8C ($n = 5$).

F   Representative gel showing levels of spliced and unspliced XBP-1 in liver tissue from control and CCl₄-induced cirrhotic mice and the effect of 4μ8C treatment.

G   miR-150 expression in liver tissue from mice in (A). *$P = 0.032$ compared to untreated controls. Ctrl ($n = 5$); Ctrl + 4μ8C ($n = 5$); CCl₄ ($n = 5$); CCl₄ + 4μ8C ($n = 5$).

H   αSMA mRNA levels in primary stellate cells isolated from mouse liver and then treated or not with TGFβ ± 4μ8C. *$P = 0.031$, $n = 3$.

I    BiP mRNA levels in primary stellate cells isolated from mouse liver and then treated with or without TGFβ ± 4μ8C. *$P = 0.008$, $n = 3$.

J   Microscopic images of primary mouse hepatic stellate cells stained for BiP (top) or αSMA (bottom). Scale bars = 20 μm.

K   Western blot showing BiP protein expression in primary mouse hepatic stellate cells.

L   Quantification of mRNA expression levels of CTGF and collagen in primary human hepatic stellate cells stimulated with TGFβ ±4μ8C.

Data information: Statistical significance was evaluated using Student's *t*-test. Significant differences are indicated with * and exact *P*-values given. Error bars indicate s.e.m.

Source data are available online for this figure.

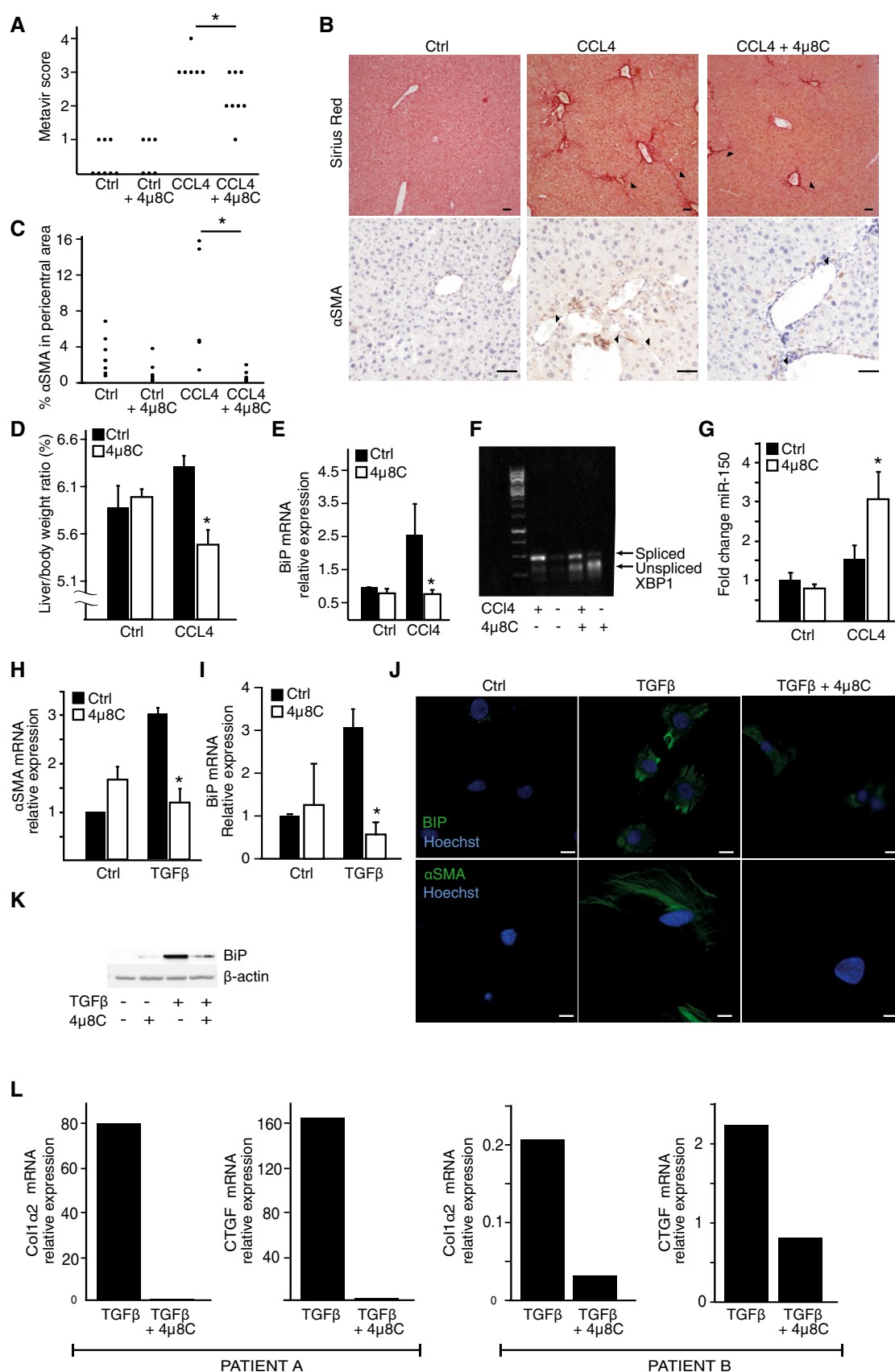

**Figure 4.**

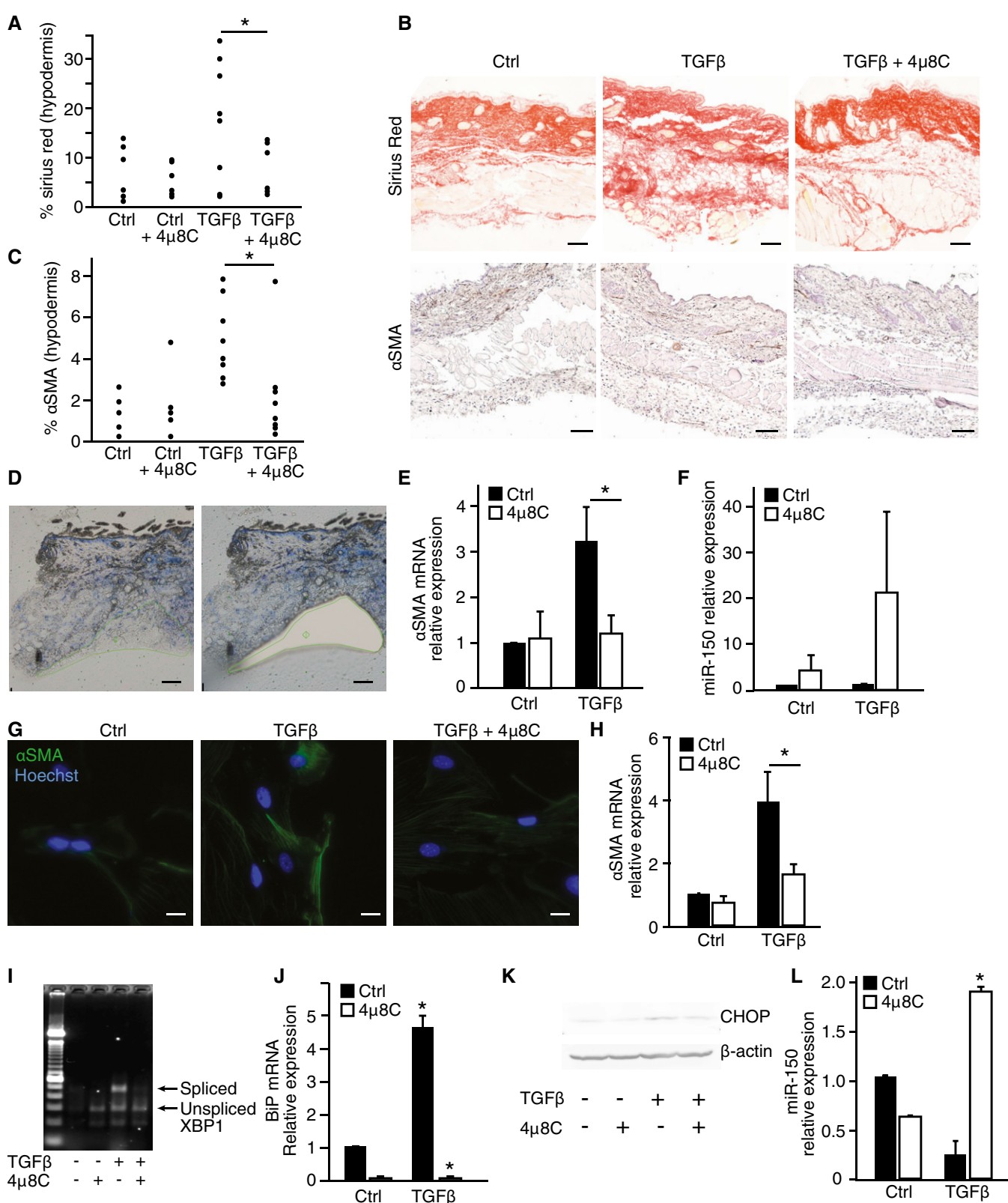

**Figure 5.**

about how ER stress influences the activation of fibroblasts to myofibroblasts. It has been shown that treatment of fibroblasts with TGFβ induces ER stress and formation of myofibroblasts, which could be inhibited with a chemical chaperone that inhibits ER stress by improving the ER protein folding capacity (Baek *et al*, 2012). In this report, we further define a direct role of

◀ **Figure 5.  Inhibition of IRE1α reduces skin fibrosis.**

A   The effect of the IRE1α inhibitor 4μ8C on TGFβ-induced skin fibrosis in C57BL/6 mice as measured by the percentage of sirius red areas in the hypodermis. Ctrl ($n$ = 6); Ctrl + 4μ8C ($n$ = 5); TGFβ ($n$ = 8); TGFβ + 4μ8C ($n$ = 8). *$P$ = 0.046.

B   Representative skin tissue sections stained with sirius red for collagen (top) and αSMA (bottom). Scale bars = 50 μm.

C   Percentage of αSMA-positive cells in the hypodermis of mice from the experimental setup in (A). *$P$ = 0.024.

D   Representative image of the area selected for laser capture microdissection (scale bars = 100 μm).

E   Expression levels of αSMA mRNA in samples of hypodermis obtained by laser capture microdissection. Ctrl ($n$ = 5), Ctrl + 4μ8C ($n$ = 5), TGFβ ($n$ = 5), TGFβ + 4μ8C ($n$ = 5). *$P$ = 0.044.

F   miR-150 expression levels in the same samples as shown in (E).

G   Immunohistochemical staining of αSMA in primary skin fibroblasts isolated from healthy mice after treatment with TGFβ ± 4μ8C. Scale bars = 20 μm.

H   αSMA mRNA expression in primary mouse skin fibroblasts after treatment with TGFβ ± 4μ8C. *$P$ = 0.049, $n$ = 4.

I   Agarose gel showing levels of spliced and unspliced XBP-1 in primary mouse skin fibroblasts treated as indicated with TGFβ and 4μ8C.

J   BiP mRNA expression in primary mouse skin fibroblasts after treatment with TGFβ in the absence and presence of 4μ8C. *$P$ = 0.00007 for TGFβ versus Ctrl and $P$ = 0.00002 for TGFβ versus TGFβ + 4μ8C. $n$ = 4.

K   Western blot showing CHOP protein expression in primary mouse skin fibroblasts treated as indicated with TGFβ and 4μ8C.

L   miR-150 mRNA levels in primary mouse skin fibroblasts treated as indicated with TGFβ and 4μ8C. *$P$ = 0.00031 compared to untreated control, $n$ = 3.

Data information: Statistical significance was evaluated using Student's *t*-test. Significant differences are indicated with * and exact *P*-values given. Error bars indicate s.e.m.

Source data are available online for this figure.

the UPR in myofibroblast differentiation as we provide evidence that inhibition of IRE1α prevents TGFβ-induced formation of myofibroblasts.

The recent finding that IRE1α is capable of degrading miRs, in addition to its well-known ability to degrade mRNA, provides a new possible connection between IRE1α and miRs that regulate the activation of myofibroblasts (Upton *et al*, 2012). IRE1α, miRs, and components of the RISC machinery are found on the cytosolic side of the rough ER membrane (Daniels *et al*, 2009; Li *et al*, 2013; Stalder *et al*, 2013), which would allow a direct interaction between IRE1α and miRs. Sequence analysis of miR-150 revealed several putative IRE1α cleavage sites (-TGCT-) (Upton *et al*, 2012) and our data demonstrate that IRE1α directly cleaves miR-150, providing a mechanism for the increased levels of miR-150 that were observed when IRE1α was inhibited in TGFβ-treated fibroblasts. Cleavage of pre-miR-150 by IRE1α probably destabilizes pre-miR-150 to prevent further processing by Dicer, as described for another ribonuclease, MPCP1 (Suzuki *et al*, 2011). Previous studies on isolated HSCs have similarly shown that TGFβ reduces miR-150 levels and that overexpression of miR-150 inhibits TGFβ-induced αSMA and collagen I and IV expression (Buck *et al*, 2000; Zheng *et al*, 2013). These results in combination with reports that miR-150 is decreased in fibrotic conditions and correlates with disease severity (Honda *et al*, 2013) indicate an anti-fibrotic function of miR-150.

One of the top target genes of miR-150 is the transcription factor c-Myb (Lin *et al*, 2008; Feng *et al*, 2014; Yang *et al*, 2015). c-Myb is known to bind the αSMA promoter and overexpression of c-Myb in HSCs increases αSMA expression, while antisense c-Myb inhibits αSMA expression (Buck *et al*, 2000). In addition, miR-150 has been shown to regulate collagen I expression by regulating the transcription factor SP1 (Zheng *et al*, 2013). These findings would suggest a pathway where fibrosis-inducing agents like TGFβ induce ER stress with activation of IRE1α. Activated IRE1α then degrades miR-150, removing the inhibitory effect of miR-150 on expression of transcription factors like c-Myb and SP1. Increased levels of these transcription factors are essential for the formation of myofibroblasts as they induce expression of αSMA and collagen I. Under normal conditions, miR-150 would continuously suppress

expression of profibrotic transcription factors and prevent formation of myofibroblasts.

A similar mechanism whereby activated IRE1α inactivates miRs that normally repress protein translation has been demonstrated for other biological responses. For example, it has previously been shown that increased caspase-2 expression and induction of apoptosis can be a result of IRE1α activation leading to destruction of miRs that under normal conditions repress caspase-2 expression (Upton *et al*, 2012). Suppression of these miRs increased caspase-2 without any additional stimuli. Further, another study showed that the thioredoxin-interacting protein (TXNIP) is increased as a result of activated IRE1α destroying miR-17 that otherwise mediates degradation of TXNIP mRNA (Lerner *et al*, 2012).

In a parallel pathway, we observed an IRE1α- and XBP-1-dependent increase in ER size when fibroblasts were stimulated with TGFβ. An enlarged ER would prepare the myofibroblast for increased production and folding of extracellular matrix proteins during wound healing and fibrosis. It has been suggested that splicing and hence activation of XBP-1 in response to ER stress promotes an increase in ER volume through increased phospholipid biosynthesis in the ER membrane (Sriburi *et al*, 2004). A functional role of XBP-1 in membrane biogenesis was originally described for differentiation of B lymphocytes into antibody-secreting plasma B cells (Reimold *et al*, 2001) and later also for ER expansion in pancreatic β-cells (Lee *et al*, 2005) and in hepatocytes (Lee *et al*, 2008).

In conclusion, our study indicates that activation of the UPR is a conserved mechanism in fibrotic pathologies in various organs and that targeting the IRE1α RNase activity with the inhibitor 4μ8C could be a potential treatment strategy for patients with progressive fibrotic diseases. We suggest that there are two mechanisms by which IRE1α contributes to myofibroblast activation (Fig 7). Firstly, the RIDD activity of IRE1α is responsible for degrading the anti-fibrotic miR-150 to allow for increased c-Myb activity leading to increased αSMA expression, thereby resulting in increased formation of myofibroblasts. Secondly, IRE1α mediates splicing of XBP-1, leading to ER expansion that contributes to the enhanced secretory capacity needed for the secretion of extracellular matrix proteins.

                                                        

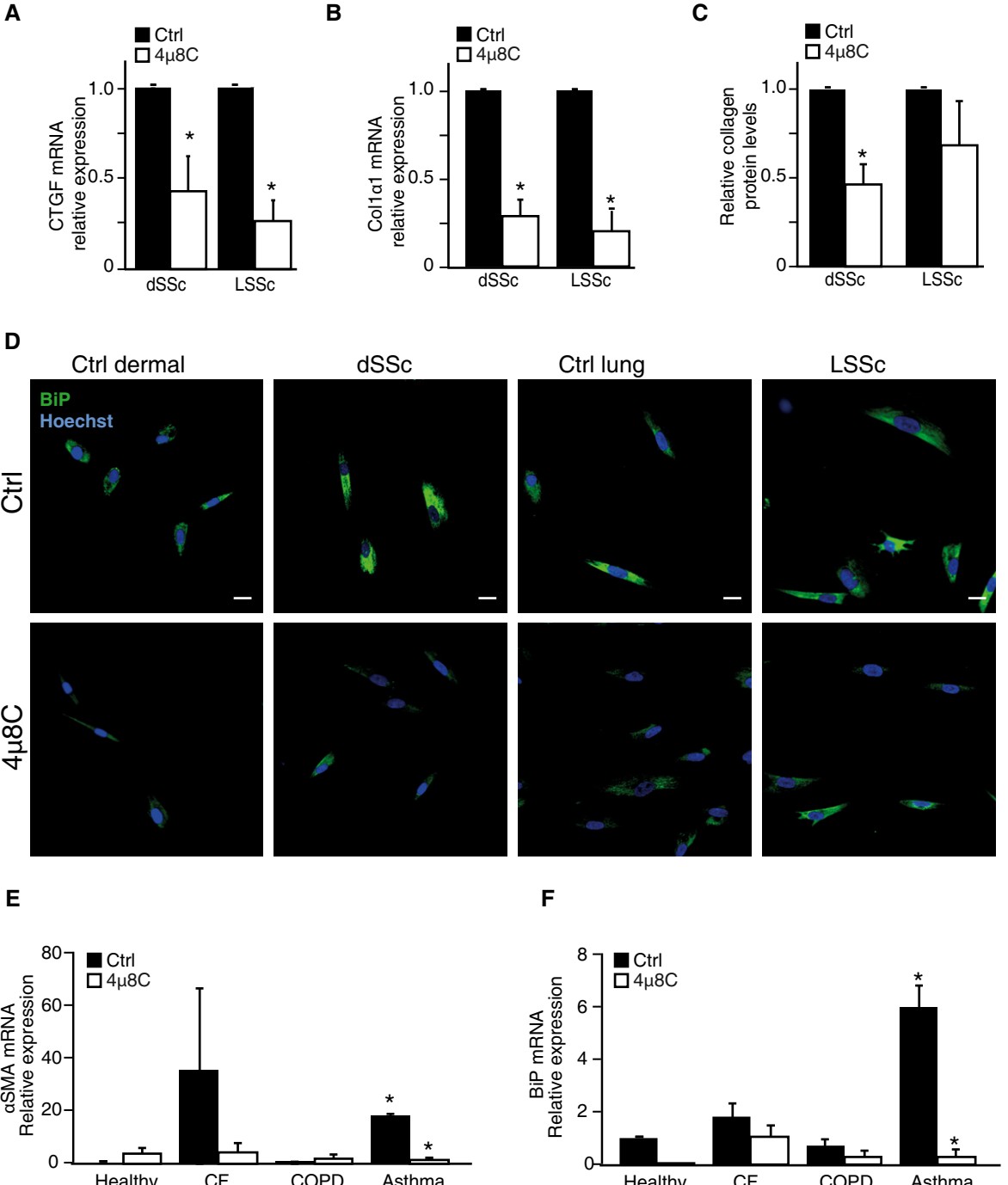

**Figure 6.** **Inhibition of IRE1α reverts the fibrotic phenotype of myofibroblasts isolated from scleroderma patients.**

A   CTGF mRNA expression levels in skin (dSSc) and lung (LSSc) fibroblasts isolated from scleroderma patients and treated with the IRE1α inhibitor 4μ8C. *P = 0.018 for dSSc (n = 6) and P = 0.003 for LSSc (n = 3).

B   Collagen 1α2 mRNA expression levels in the same samples as in (A). *P = 0.0030 for dSSc and P = 0.0036 for LSSc.

C   Quantification of collagen protein levels in the same samples as in (A). *P = 0.010.

D   Immunohistochemical staining to detect BiP protein in fibroblasts isolated from scleroderma patients treated or not with 4μ8C. Scale bars = 20 μm.

E   mRNA levels of αSMA in primary lung fibroblasts isolated from patients with either cystic fibrosis (CF), chronic obstructive pulmonary disease (COPD), or asthma. *P = 0.00001 for asthma versus healthy controls, and P = 0.00003 for ctrl versus 4μ8C in asthma patients.

F   Quantification of mRNA levels for BiP in same samples as in (E). *P = 0.0023 between asthma and healthy controls and P = 0.0031 between 4μ8C-treated and control treatment in fibroblasts from asthma patients.

Data information: Statistical significance was evaluated using Student's *t*-test. Significant differences are indicated with * and exact *P*-values given. Error bars indicate s.e.m. *n* = 3 in (E, F).

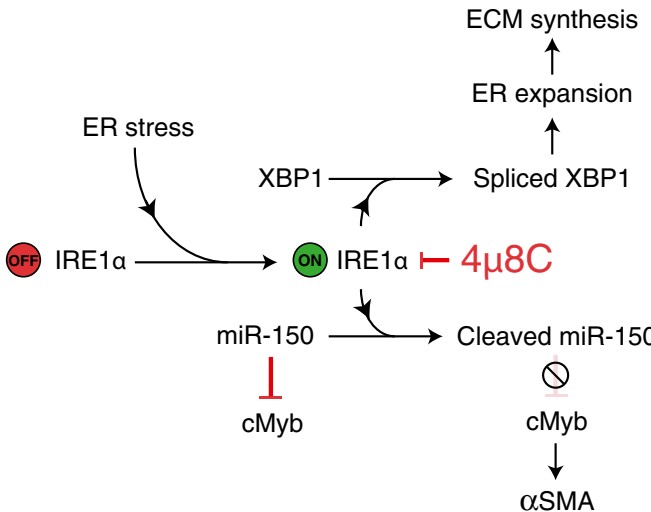

**Figure 7. Model for the role of IRE1α in myofibroblast activation.**

In the absence of ER stress, miR-150 inhibits c-Myb expression to prevent c-Myb-induced αSMA expression. ER stress, for example, as a result of fibrotic stimuli, activates IRE1α that directly cleaves and inactivates miR-150 resulting in increased c-Myb expression and as a consequence increased expression of αSMA and activation of myofibroblasts. In addition, IRE1α-mediated splicing of XBP-1 mRNA leads to ER expansion and enhanced secretory capacity of extracellular matrix proteins. Inhibition of the endoribonuclease activity of IRE1α by 4μ8C will lead to increased miR-150 levels followed by reduced c-Myb and αSMA levels. Treatment with 4μ8C will also reduce XBP-1 splicing, ER expansion, and secretion of matrix proteins.

# Materials and Methods

## Cell culture and reagents

Human fetal lung fibroblast (HFL1, ECACC) was routinely cultured in MEM supplemented with nonessential amino acids, 25 mM HEPES (Sigma), 10% FBS (Gibco), 2 mM glutamax (Gibco), and 1 mM sodium pyruvate (Gibco). Cells were cultured at 37°C with 5% $CO_2$. For starvation, stimulation and migration growth medium without FBS was used.

Primary cells were isolated from skin and lung, using the tissue explant outgrowth method. Shortly, fresh tissue was obtained postmortem from adult mice, rinsed in PBS, cut in 5-$mm^2$ pieces, and subsequently added to culture plates with medium (DMEM, 10% FCS, penicillin, streptomycin, and fungizone) to allow outgrowth of fibroblasts. Mouse hepatic stellate cells were isolated using a pronase/collagenase digestion, followed by Nycodenz gradient centrifugation (Maschmeyer *et al*, 2011). Human hepatic stellate cells were isolated as previously described (Rombouts *et al*, 2012). Fibroblasts were explanted and cultured from skin or lung tissue taken from patients with scleroderma or from normal control subjects (kind gift of Prof Chris Denton, Royal Free Hospital UK). The patients fulfilled the American College of Rheumatology criteria. All samples were obtained after informed consent and under institutional local research ethics committee approval (Royal Free and Medical School Local Research Ethics Committee) (Ponticos *et al*, 2015). Each of the primary cell lines in this study is derived from a different individual or mouse.

Mouse embryonic fibroblasts (MEFs) either WT or deficient for IRE1α (kind gift from R.J. Kaufman, Sanford-Burnham Institute) were grown in DMEM media supplemented with 10% fetal bovine serum. Primary lung fibroblasts from patients with asthma (194912, Lonza), COPD (195277, Lonza), and CF (194843, Lonza) and healthy controls (CC-2512; Lonza) were grown in FGM-2 medium (CC3131 and CC4134, Lonza).

For *in vitro* experiments, cells were detached using trypsin–EDTA (Gibco), resuspended in growth medium, and plated at a density of $5 \times 10^3$ cells/$cm^2$. Cells were allowed to attach and left undisturbed for 8 h before being starved for 16 h. After that time, fresh starvation medium containing indicated growth factors or substances was added. Cells were exposed for 48 h to 5 ng/ml TGFβ and/or 100 μM 4μ8C, unless stated otherwise. Previous studies have shown that concentrations up to 128 μM of 4μ8C cause no measurable toxicity and that a concentration of 100 μM leads to a complete inhibition of XBP-1 splicing (Cross *et al*, 2012). Actinomycin D (A1410, Sigma) was added to the cells at a concentration of 5 μg/ml.

## miR cleavage assay

Pre-miR-150 was synthesized by Thermo Fisher Scientific miRIDIAN using the published sequence in the miRBase (www.mirbase.org). The dephosphorylated pre-miRNA was end-labeled with $^{32}$P-ATP and incubated with 2.5 μg of recombinant IRE1α for 6 h at 37°C in a reaction buffer (20 mM Hepes, 50 mM NaCl, 1 mM DTT, 10 mM ATP). Cleaved products were resolved using a 15% denaturing polyacrylamide gel. The dried gel was exposed to an autoradiographic film overnight. The XBP-1 mini-stem loop was synthesized by Integrated DNA Technologies and processed similarlyto pre-miR-150. The pre-miR-150 and XBP-1 mini-stem-loop sequences are provided in Table EV1.

## Quantitative RT–PCR of mRNA or miRNA and XBP-1 splicing assay

RNA was isolated from tissue or cell culture using the Qiagen miRNeasy Mini kit according to the manufacturer's instructions. Briefly, 500 ng of miRNA/mRNA was reverse-transcribed using miScript II RT Kit, and amplifications performed using primers summarized in Table EV2. Normalization was done using *ActB/RPS18/TBP* for mRNA and *SNORD68/SNORD95* for miRNA. The spliced form of XBP-1 was detected by incubation of the semi-quantitative PCR product with PstI enzyme for 3 h at 37°C (Calfon *et al*, 2002). The upper, undigested product corresponds to XBP-1 spliced variant as seen following migration in a 1.5% agarose gel.

## Transfections and nucleofections

HFL1 cells were transiently transfected with 300 ng of IRE1α WT, K599A or K907A, and miR-150 expression plasmid (in courtesy of D. Bartel, Addgene #26310) using Lipofectamine LTX and the PLUS reagent for 24 h. Nucleofection of siXBP-1 (sc-38627, Santa Cruz), sic-Myb (4392420, Life Technologies), si-miR-150 (219300, Qiagen), or siCtrl (100 pmol) was done using Amaxa Nucleofector program S-05 in Ingenio electroporation solution (Mirus Bio LLC). Cells were starved overnight and subsequently treated with 5 ng/ml TGFβ and/or 4μ8C (100 μM) for 48 h.

## Animal experiments

Eight-week-old, male C57Bl/6 mice were obtained from Scanbur (Denmark). Mice were housed in standard conditions with normal dark–light cycle. Animals were acclimatized to their environment for 1 week and given *ad libitum* access to water and food throughout the experiments. Mice were randomly divided into different treatment groups prior to starting the experiment. During experiments, mice were weighted twice per week and their health was evaluated using the Karolinska Institute health monitoring template, whereby a total score of > 0.4 or a single score of > 0.3 was considered a humane endpoint for euthanasia. Sample size was determined from previous experience (Geerts *et al*, 2008; Van de Veire *et al*, 2010; Van Steenkiste *et al*, 2011).

## Skin fibrosis model

Mini-osmotic pumps (Alzet 1003D, 1 µl/h) containing TGFβ (200 ng in 0.1% BSA in saline) or controls (0.1% BSA in saline) were implanted subcutaneous in 32 mice under isoflurane (Forane) anesthesia. Postoperative analgesia (Temgesic) was provided daily for 3 days. Mice were subcutaneously injected every day near the implantation site with 10 µg/g 4µ8C in saline or 0.08% DMSO in saline (control). Skin biopsies were collected after 3 days using 4-mm biopsy punches.

## Liver cirrhosis model

Sixteen mice were injected twice per week with $CCl_4$ (Sigma) (1:1 dissolved in olive oil; 1 mg/kg) and 5% alcohol was added to drinking water. Similarly, 16 control mice received physiological saline solution (1 ml/kg) and no alcohol was added. Mice were injected twice per week with 10 µg/g 4µ8C in saline or 0.08% DMSO in saline (control). After 15 weeks, mice were euthanized and samples were taken from the liver.

## Stainings and immunohistochemistry

Tissue samples were fixed in 4% paraformaldehyde for 24 h and subsequently embedded in paraffin. 5-µm slides were cut and dried overnight. Sections were de-paraffinized and rehydrated prior to staining. Collagen was stained using the picrosirius red staining with an incubation time of 30 min, followed by 10-min washing in distilled water. For the immunohistochemical staining of αSMA, an antigen retrieval was done at 95°C in sodium citrate buffer, endogenous peroxidase activity was blocked using 3% $H_2O_2$, and endogenous mouse IgG was blocked using a mouse-on-mouse staining kit (ab127055, Abcam) prior to a 1-h incubation with the primary antibody (A2547, Sigma) followed by a HRP-coupled secondary antibody using DAB as substrate. Slides were washed in TBS-T (0.1% Triton X-100) 3 × 10 min after primary and secondary antibody incubations. Slides were counterstained with Mayer's hematoxylin. For the immunofluorescent stainings of skin and liver tissue, antigen retrieval was done at 95°C in sodium citrate buffer and endogenous mouse IgG was blocked using a rodent blocking buffer (ab127055, Abcam) following the manufacturer's protocol. Samples were incubated overnight at 4°C with primary antibodies against αSMA (A2547, Sigma), BiP (ab21685,

Abcam), and/or CHOP (sc575, Santa Cruz). A 20-min incubation was used for the secondary antibody (Alexa Fluor 488 and Alexa Fluor 555) and cell nuclei were stained with Hoechst for 5 min.

For staining of cells in culture, cells were fixed for 10 min in 4% paraformaldehyde, washed with PBS, and blocked for 30 min using 1% BSA in PBS + 0.1% Tween, followed by a 1-h incubation with antibodies against αSMA (clone 1A4, Sigma) or BiP (ab21685). A 20-min incubation was used for the secondary antibody (Alexa Fluor 488) and cell nuclei were stained with Hoechst staining for 5 min. Images were acquired using a Nikon eclipse 90i microscope equipped with a DS-Qi1Mc camera and Nikon Plan Apo objectives. NIS-Elements AR 3.2 software was used to save and export the images. The different channels (RGB) of immunofluorescent images were merged using ImageJ software. Quantifications were performed blindly with ImageJ software by conversion to binary images after color de-convolution to separate DAB staining, as previously described (Ruifrok & Johnston, 2001).

## Collagen measurements

Cell culture medium was collected after 48 h of stimulation and the concentration of soluble collagen was measured using the Sircol Assay kit (BCS1000, Nordic BioSite, Täby, Sweden) following manufacturer's protocol.

## Immunoblotting

Samples were lysed and protein concentrations were determined using the BCA reagent. Equal amounts of proteins were analyzed by standard SDS–PAGE technique. Membranes were incubated with the respective antibodies (see Table EV2) and analyzed using the Li-Cor Odyssey imaging system.

## Laser microdissection

Mouse skin samples were embedded in OCT and cut into 8-µm sections and mounted on frame slides (POL-Membrane 0.9 µm, Leica Microsystems, Wetzlar, Germany). Sections were fixed in acetone, washed with RNase-free water, then stained with RNase-free hematoxylin (Arcturus® HistoGene® Staining Solution, Applied Biosystems, Foster City, CA, USA) for 1 min, dehydrated, and dried. The hypodermis layer was laser microdissected using a Leica LMD6000 B microscope. Quantification of mRNA or miRNA expression in the microdissected layers was performed as previously described.

## Quantification of ER expansion

For ER staining, cells were washed once with HBSS and incubated for 30 min with 1 µm of ER-Tracker Red (Invitrogen). After 5-min fixation with 3.7% formaldehyde at 37°C, cells were counterstained with Hoechst and visualized using a Zeiss LSM 700 confocal microscope. Quantification of ER morphology was done blindly using ImageJ software. For the electron microscopy images, cells were fixed in 2% paraformaldehyde with 0.25% glutaraldehyde in PBS, postfixed in 1% $OsO_4$, and embedded in Epon (Ladd Research Industries, Burlington, VT, USA). The material was studied at 100 kV in a Jeol 1230 electron microscope (Jeol, Akishima, Tokyo, Japan). Electron micrographs were taken with a Gatan multiscan

**The paper explained**

**Problem**

Fibrosis is characterized by an excessive accumulation of fibrous connective tissue that typically occurs in response to chronic inflammation. Despite that fibrosis can affect nearly every tissue in the body and is a major cause of morbidity and mortality in a variety of diseases, there are few treatments that specifically target the pathogenesis of fibrosis.

**Results**

We provide evidence that the ER stress sensor IRE1α mediates fibrosis by directly cleaving and inactivating miR-150. Under normal conditions, miR-150 represses c-Myb expression and thereby c-Myb-induced αSMA expression in fibroblasts which hinders formation of activated myofibroblasts. In addition, we find that IRE1α-mediated splicing of XBP-1 is required for expansion of the ER in activated myofibroblasts, which could be a mechanism that supports the enhanced secretion of extracellular matrix proteins. Inhibition of IRE1α, using the inhibitor 4μ8C, was found to inhibit TGFβ-induced activation of myofibroblasts *in vitro*, reduced liver and skin fibrosis *in vivo*, and reverted the profibrotic phenotype of activated myofibroblasts isolated from patients with systemic sclerosis.

**Impact**

Our results indicate that ER stress could be an important and conserved mechanism in the pathogenesis of fibrosis and that targeting components of the ER stress pathway might be therapeutically relevant for treating patients with fibrotic diseases.

camera model 791 with Gatan digital micrograph software version 3.6.4 (Gatan, Pleasanton, CA, USA).

## Statistics

Data are presented as mean ± s.e.m. An unpaired, two-tailed Student's *t*-test was used to compare different groups; a $P < 0.05$ was considered statistically different.

## Study approval

The ethical committee for animal studies approved all methods involving animals (C45/13 and C95/14).

**Expanded View** for this article is available online.

## Acknowledgements

We thank Dr Paul O'Callaghan for valuable input during preparation of the manuscript. FH is funded by the Wenner-Gren Foundation and Cancerfonden. PG is supported by Barncancerfonden, Cancerfonden, and Lions Cancer Research Fund in Uppsala. JK is supported by Cancerfonden. FB is supported by Fonds de recherche du Québec-Santé.

## Author contributions

FH was involved in the study design, animal experiments, and *in vitro* work. FB contributed to the study design and the *in vitro* experiments. MP was involved in the experiments with fibroblasts from scleroderma patients. KR provided hepatic stellate cells from patients. JL assisted with the laser capture microdissection. JK and PG were involved with the study design and project management. All authors contributed to writing of the manuscript.

## Conflict of interest

The authors declare that they have no conflict of interest.

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
