## [Review Process File · EMBO Molecular Medicine]

Endoplasmic reticulum stress enhances fibrosis through IRE1-mediated degradation of miR-150 and XBP-1 splicing

Femke Heindryckx, François Binet, Markella Pontico, Krista Rombouts, Joey Lau, Johan Kreuger, Pär Gerwins

Corresponding author: Pär Gerwins, Uppsala University

Review timeline:

Submission date:	08 October 2015
Editorial Decision:	03 November 2015
Revision received:	06 March 2016
Editorial Decision:	23 March 2016
Revision received:	16 April 2016
Accepted:	20 April 2016

Transaction Report:

Editor: Roberto Buccione

1st Editorial Decision

03 November 2015

Thank you for the submission of your manuscript to EMBO Molecular Medicine. We have now heard back from the three Reviewers whom we asked to evaluate your manuscript.

Although the Reviewers are globally positive and agree on the potential interest of the manuscript, a few issues are raised that require your action. I will not dwell into much detail, but I would like to highlight the main points.

Reviewer 1 expresses two main concerns. On one hand, s/he would like you to validate the main conclusions by direct experimentation on primary myofibroblasts and cells from CCL4-treated mice. On the other, the reviewer notes that there is no direct evidence that indeed XBP1 splicing is required for TGFbeta-induced activation. This Reviewer also lists other items for your attention.

Reviewer 2, in connection with the fact that you suggest that miR-150 is cleaved by RIDD, would like you to establish whether it is associated with the ER membrane and would like you to clarify experimentally whether the stability of miR-150 is reduced in fibrosis. This reviewer would also like you to better discuss the rationale for IRE-1 cleavage of miR-150.

Reviewer 3 is also not convinced of the causal connection between IRE-1 and miR-150 and notes important missing key experimental evidence required to support your claims.

In conclusion, while publication of the paper cannot be considered at this stage, given the potential interest of your findings and after internal discussion, we have decided to give you the opportunity to address the criticisms.

We are thus prepared to consider a revised submission, with the understanding that the Reviewers' concerns must be addressed, especially in terms of firmly establishing the causal connections with additional experimental data where appropriate and that acceptance of the manuscript will entail a second round of review. The overall aim is to significantly upgrade the relevance and conclusiveness of the dataset, which of course is of paramount importance for our title.

Please note that it is EMBO Molecular Medicine policy to allow a single round of revision only and that, therefore, acceptance or rejection of the manuscript will depend on the completeness of your responses included in the next, final version of the manuscript.

EMBO Molecular Medicine now requires a complete author checklist (<http://embomolmed.embopress.org/authorguide>) to be submitted with all revised manuscripts. Provision of the author checklist is mandatory at revision stage; The checklist is designed to enhance and standardize reporting of key information in research papers and to support reanalysis and repetition of experiments by the community. The list covers key information for figure panels and captions and focuses on statistics, the reporting of reagents, animal models and human subject-derived data, as well as guidance to optimise data accessibility. This checklist especially relevant in this case given the issues raised with respect to statistical treatment and animal numbers.

As you know, EMBO Molecular Medicine has a "scooping protection" policy, whereby similar findings that are published by others during review or revision are not a criterion for rejection. However, I do ask you to get in touch with us after three months if you have not completed your revision, to update us on the status. Please also contact us as soon as possible if similar work is published elsewhere.

I look forward to seeing a revised form of your manuscript as soon as possible.

***** Reviewer's comments *****

Referee #1 (Remarks):

The authors demonstrate that IRE1 α , which mediates one of the three canonical arms of the unfolded protein response, is critical for fibrosis development. The authors do an elegant job of showing the role of IRE-1 α in fibrogenesis within several organ systems and highlighting two mechanisms through which IRE1 α promotes HSC activation. The regulation of miR150 by IRE1 is a novel finding that provides a mechanism for how ER stress leads to activation of myofibroblasts. Overall, the paper clearly shows an important role for IRE1 in myofibroblast activation and highlights therapeutic potential for the targeting of IRE1. Overall role of ER stress in myofibroblast activation is timely and topical.

MAJOR CRITICISMS

1. The authors state that their impact revolves around ER stress as a conserved mechanism in the

pathogenesis of fibrosis; however, they omit analysis of the UPR in isolated myofibroblasts from liver or skin. It would strengthen the paper to observe increased ER stress upon activation of primary myofibroblasts with TGF β and in cells isolated from mice with CCl₄. The authors show increased BiP mRNA expression from whole liver, but ER stress in hepatocytes is common in fibrosis models, thus it is not indicative of ER stress in myofibroblasts. Thus, greater attention is needed on isolated cells to corroborate whole tissue analyses.

2. In the results section describing Figure 3, the authors hypothesize that, along with ER expansion, XBP1 splicing is at least partially required for TGF β -induced activation; however there are no experiments testing this. The authors should analyze XBP1 and myofibroblast activation directly.

MINOR CRITICISMS

1. It is unclear how collagen content was quantified in Figure 1F and 6C. Neither the body of the paper nor the figure legends indicate the method used.

2. In the first paragraph, Figure 1E is referred to as Figure 1B.

3. In Figure 1D it is difficult to see the fourth lane - is this typical of co-treatment with TGF β and 4 μ 8C? Need to have a clear gel.

4. With the exception of ER expansion, the authors rely entirely on mRNA expression/splicing to describe the induction of ER stress. Immunoblotting to examine BiP and CHOP expression is needed in Figure 1, 4, and 6

5. The figure legend from Figure 6 reads "Myofibroblasts isolated from scleroderma patients exhibit markers of ER stress...." but there are no markers of ER stress analyzed.

Referee #2 (Remarks):

Fibrosis is caused by an excessive accumulation of fibrous materials including collagens and other extracellular matrix proteins by activated myoblasts, which express alpha smooth muscle actin (alpha-SMA). It has been reported that miR-150 levels were reduced upon fibrosis, and that overexpression of miR-150 inhibits expression of alpha-SMA and collagen during fibrosis. In addition, a transcription factor c-Myb is one of targets of miR-150

The unfolded protein response (also called the endoplasmic reticulum (ER) stress response) has been implicated in the development of fibrosis. The mammalian unfolded protein response consists of three response pathways, that is, the ATF6, IRE1 and PERK pathways. IRE1 is a sensor molecule located in the ER membrane, and activated by ER stress (accumulation of unfolded proteins in the ER). Activated IRE1 converts XBP1 pre-mRNA to mature mRNA by cytoplasmic splicing, from which an active transcription factor pXBP1(S) is translated, leading to expansion of ER. Activated IRE1 cleaves mRNAs associated with ER membrane by the mechanism of RIDD.

In this manuscript, the authors revealed that IRE1 is activated during fibrosis, and that pharmacological inhibition of IRE1 by 4mu8C or kinase-dead IRE1 mutants reduced expression of

collagen and alpha-SMA. Interestingly, expression of miR-150 was decreased by activated IRE1. The authors found IRE1 recognition / cleavage sites in miR-150. They also found that miR-150 reduced cMyb expression, and claimed that IRE1 is activated and cleaved miR-150 by RIDD during fibrosis, which results in stabilization of c-Myb mRNA and transcriptional activation of alpha-SMA and collagens by c-Myb, leading to fibrosis. Moreover, the authors showed that IRE1 enhances ER expansion, which supports expression of lots of collagen and enhances onset of fibrosis. From these observations, the authors concluded that "endoplasmic reticulum stress enhances fibrosis through miR-150 degradation and XBP-1 splicing".

Since data presented here are all clear and ample, and the subject seems to attract interest of readers, the reviewer thinks that the manuscript would be suitable for publication in the journal of EMBO Molecular Medicine.

<Critiques>

(1) The reviewer wonders for what purpose IRE1 cleaves miR-150. What is the biological benefit? According to the author's conclusion, whenever IRE1 is activated, miR-150 is cleaved and fibrosis is induced. Why do mammalian cells conserve such a dangerous mechanism? The authors should explain about this.

(2) The author claimed that miR-150 is cleaved by RIDD. The reviewer wonders if miR-150 is associated with ER membrane, since most of RIDD substrates are associated with ER membrane (because IRE1 is localized at the ER membrane). Is it possible to show that miR-150 is associated with ER membrane, or are there any papers that revealed ER-localization of miR-150?

(3) Figure 2F and 2G: the consensus of IRE1 recognition site is C-G-G motif in the loop structure that consists of seven nucleotides. But the putative IRE1 recognition sequence of miR-150 is not similar to this consensus. In addition, cleavage efficiency of miR-150 by IRE1 is considerably low as compared with that of XBP1 mRNA. The authors should show that stability of miR-150 is reduced and transcription of miR-150 does not change during fibrosis.

<Typographical errors>

(1) Page 3, line 12: "in heart failure.." should be "in heart failure."

Referee #3 (Comments on Novelty/Model System):

The manuscript included data from in vitro, in vivo and human sample experiments which are technically well done. They also use a combination of genetic and pharmacologic strategies, which is also a strength.

Referee #3 (Remarks):

In this paper the authors' report the role of Endoplasmic Stress in tissue fibrosis. Specifically they show that inhibiting IRE-1 alpha pharmacologically, blocks TGF-beta activation of myofibroblasts and prevents liver and skin fibrosis in mouse models. They supplemented this data by using genetic strategies, using IRE-1aa KO and IRE1a mutants lacking endoribonuclease activity, to confirm the role IRE-1a in myofibroblast differentiation. Further they show data that suggests IRE-1a mediates its effects via its ribonuclease activity on miR-150 and through an XBP-1-dependent pathway. They

conclude that targeting ER stress may be a viable therapeutic approach for tissue fibrosis.

General Comment: Fibrosis of tissues is a common problem for which there is incomplete understanding and limited treatment options. For these reasons the work reported in this paper is important. There has been increased interest in exploring the role of ER stress in tissue fibrosis and this paper provides further evidence for its importance. The paper is technically well done as it includes both genetic and pharmacologic data and has in vitro, in vivo and human data. The authors provide substantial evidence linking ER stress as a mediator of TGF- β induced myofibroblast differentiation and tissue fibrosis. The evidence that IRE-1a activation is important for these effects is compelling and relatively novel. The data that IRE-1a mediates its effects thru miR-150 and ER expansion is less well-supported. While the authors show data that support these pathways, they lack key experiments which would prove IRE-1a is mediating its effects via miR-150 and c-MYB, or how ER volume is increasing. The use of two tissue fibrosis models and use of human fibroblast to establish biologic relevance is a strength of the paper. Since they used lung fibroblasts, use of a lung fibrosis model would strengthen the generalizability of the paper.

Specific Comments:

- In figure 2C, there is no difference in miR-150 in response to TGF- β in the IRE-1a KO cells. This discrepancy in response compared to the experiments in 2A and 2B is not discussed and needs clarification.
- While it is clear from the data that inhibiting IRE-1a leads to increased mir150 expression, the authors have not provided conclusive data IRB-1a is mediating its fibrotic effects via miR150. For example does inhibiting mir150 in in IRE-1a KO cells rescue aSMA expression?
- There is no blot confirming overexpression of miR150 (Figure 2D).
- The blot showing inhibition of TGF- β induced aSMA expression in miR150 overexpressing cells is not very convincing.
- There is no experiment that links mir150 inhibited aSMA expression directly to its effects on cMYB. This is a key experiment in order to establish the mechanism by which mir150 is mediating its anti-fibrotic effects.

1st Revision - authors' response

06 March 2016

Response to reviewer's comments on manuscript EMM-2015-05925

We would like to thank the editors and reviewers for their careful evaluation of our manuscript and for providing constructive criticism and suggestions. We have tried to address all the points raised by the reviewers in the revised manuscript, including new experimental work that strengthens our findings and improves the quality of this manuscript.

Referee #1 (Remarks):

The authors demonstrate that IRE1a; which mediates one of the three canonical arms of the unfolded protein response, is critical for fibrosis development. The authors do an elegant job of showing the role of IRE-1a in fibrogenesis within several organ systems and highlighting two mechanisms through which IRE1a promotes HSC activation. The regulation of miR150 by IRE1 is a novel finding that provides a mechanism for how ER stress leads to activation of myofibroblasts. Overall, the paper clearly shows an important role for IRE1 in myofibroblast activation and highlights therapeutic potential for the targeting of IRE1. Overall role of ER stress in myofibroblast activation is timely and topical.

MAJOR CRITICISMS

1. The authors state that their impact revolves around ER stress as a conserved mechanism in the pathogenesis of fibrosis; however, they omit analysis of the UPR in isolated myofibroblasts from liver or skin. It would strengthen the paper to observe increased ER stress upon activation of primary myofibroblasts with TGF β ; and in cells isolated from mice with CCl₄. The authors show increased BiP mRNA expression from whole liver, but ER stress in hepatocytes is common in fibrosis models, thus it is not indicative of ER stress in myofibroblasts. Thus, greater attention is needed on isolated cells to corroborate whole tissue analyses.

We agree with the comment that ER-stress could contribute to fibrosis through other cell types than myofibroblasts. Specifically in the liver, it is not uncommon that hepatocytes also express increased ER-stress markers and that this could actively contribute to the disease progression¹⁻³.

To address this concern we have performed immunofluorescent co-stainings on liver and skin sections, which show co-localisation between the myofibroblast marker α SMA and the ER-stress markers BiP and CHOP (Fig EV2 and EV3) and these experiments suggest that myofibroblasts are experiencing increased ER-stress. As suggested by the referee, we also measured BiP mRNA expression (Fig 4I and Fig 5J), XBP1 splicing (Figure 5I) and BIP/CHOP protein levels (Fig 4J,K and Figure 5K) in hepatic stellate cells and skin fibroblasts isolated from mice.

2. In the results section describing Figure 3, the authors hypothesize that, along with ER expansion, XBP1 splicing is at least partially required for TGF β -induced activation; however there are no experiments testing this. The authors should analyze XBP1 and myofibroblast activation directly.

We agree that this important information was missing in the original manuscript. Fibroblasts were therefore transfected with siRNA targeting XBP-1 and secreted collagen measured after TGF β treatment. While collagen secretion significantly increased in the untransfected controls, this was not the case after siXBP-1 transfection (Figure 3D), thus supporting the hypothesis that XBP-1 is at least in part contributing to TGF β -induced activation of myofibroblasts.

MINOR CRITICISMS

1. It is unclear how collagen content was quantified in Figure 1F and 6C.

Neither the body of the paper nor the figure legends indicate the method used.

Collagen was measured using the Sircol Assay, and this information has accordingly been added to the methods section.

2. In the first paragraph, Figure 1E is referred to as Figure 1B.

This mistake has been corrected.

3. In Figure 1D it is difficult to see the fourth lane - is this typical of co-treatment with TGF β ; and 4u8c? Need to have a clear gel.

We have tried to improved images of gels used for detection of spliced/unspliced XBP-1 (Fig 4F and Fig 5I). We agree that it is not always clear to see the spliced/unspliced XBP1, specifically in conditions where XBP1 splicing was inhibited using 4u8C.

4. With the exception of ER expansion, the authors rely entirely on mRNA expression/splicing to describe the induction of ER stress. Immunoblotting to examine BiP and CHOP expression is needed in Figure 1, 4, and 6

We have added both Western Blots (Figure 1D, Figure 4K and Figure 5K) as well as immunohistochemical stainings (EV Figure 2 and 3; Figure 6D) to show ER-stress activation at the protein level.

5. The figure legend from Figure 6 reads "Myofibroblasts isolated from scleroderma patients exhibit markers of ER stress...." but there are no markers of ER stress analyzed.

This has been corrected in the revised version of the manuscript.

Referee #2 (Remarks):

Fibrosis is caused by an excessive accumulation of fibrous materials including collagens and other extracellular matrix proteins by activated myoblasts, which express a smooth muscle actin (α SMA). It has been reported that miR-150 levels were reduced upon fibrosis, and that overexpression of miR-150 inhibits expression of α SMA and collagen during fibrosis. In addition, a transcription factor c-Myb is one of targets of miR-150

The unfolded protein response (also called the endoplasmic reticulum (ER) stress response) has been implicated in the development of fibrosis. The mammalian unfolded protein response consists of three response pathways, that is, the ATF6, IRE1 and PERK pathways. IRE1 is a sensor molecule located in the ER membrane, and activated by ER stress (accumulation of unfolded proteins in the ER). Activated IRE1 converts XBP1 pre-mRNA to mature mRNA by cytoplasmic splicing, from which an active transcription factor pXBP1(S) is translated, leading to expansion of ER. Activated IRE1 cleaves mRNAs associated with ER membrane by the mechanism of RIDD.

In this manuscript, the authors revealed that IRE1 is activated during fibrosis, and that pharmacological inhibition of IRE1 by 4u8C or kinase-dead IRE1 mutants reduced expression of collagen and α SMA. Interestingly, expression of miR-150 was decreased by activated IRE1. The authors found IRE1 recognition / cleavage sites in miR-150. They also found that miR-150 reduced cMyb expression, and claimed that IRE1 is activated and cleaved miR-150 by RIDD during fibrosis, which results in stabilization of c-Myb mRNA and transcriptional activation of α SMA and collagens by c-Myb, leading to fibrosis. Moreover, the authors showed that IRE1 enhances ER expansion, which supports expression of lots of collagen and enhances onset of fibrosis. From these observations, the authors concluded that "endoplasmic reticulum stress enhances fibrosis through miR-150 degradation and XBP-1 splicing".

Since data presented here are all clear and ample, and the subject seems to attract interest of readers, the reviewer thinks that the manuscript would be suitable for publication in the journal

of *EMBO Molecular Medicine*.

<Critiques>

(1) The reviewer wonders for what purpose IRE1 cleaves miR-150. What is the biological benefit? According to the author's conclusion, whenever IRE1 is activated, miR-150 is cleaved and fibrosis is induced. Why do mammalian cells conserve such a dangerous mechanism? The authors should explain about this.

This is a very interesting question. One hypothesis for the biological relevance of this mechanism is that fibroblast activation under certain conditions (such as wound healing or acute injuries) requires a fast response of resident fibroblasts to differentiate to contractile myofibroblasts. Cleaving miR150 could be a rapid way to increase fibroblast activation in these acute situations.

In addition, when myofibroblasts are activated (by TGF β , other growth factors or stress), this can lead to the induction of the UPR, thereby adapting to the increased need of protein translation. ER-stress in itself will further push myofibroblast activation, thus causing a positive feedback loop. This positive feedback mechanism could be a way to prevent uncontrolled fibroblast activation, since severe ER-stress will induce apoptosis^{4, 5}. Elimination of myofibroblasts by apoptosis is essential during normal cutaneous wound healing⁶, but seems to be disrupted in fibrotic disorders⁷⁻⁹. This could lead to the continuous induction of the UPR and uncontrolled activation of myofibroblast seen in fibrosis.

(2) The author claimed that miR-150 is cleaved by RIDD. The reviewer wonders if miR-150 is associated with ER membrane, since most of RIDD substrates are associated with ER membrane (because IRE1 is localized at the ER membrane). Is it possible to show that miR-150 is associated with ER membrane, or are there any papers that revealed ER-localization of miR-150?

We thank the reviewer for bringing this to our attention. The RISC complex, which comprises Dicer-Ago2-miRNA, has been localized to the rough endoplasmic reticulum (rER) or perinuclear compartment in several papers¹⁰⁻¹². Moreover, the RISC complex is found on the cytosolic side but not luminal side of the rER¹⁰, similar to IRE1 α endoribonuclease activity¹³. We have now as suggested included this information in the manuscript.

To our knowledge, not much is known about the exact intracellular localization of miR-150 specifically. In situ detection of miR150 precursors and mature miR150, showed both cellular and nuclear localization in the cancer cell lines Jurkat and HL60¹⁴. miR-150 is often secreted by cells and found in the bloodstream as a circulating miRNA¹⁵⁻¹⁷. Recent studies have shown that argonaute 2 forms a protein-miRNA complex with miR-150 that allows its extracellular transport¹⁸. Interestingly, argonaute 2 is also associated with the ER-membrane.

(3) Figure 2F and 2G: the consensus of IRE1 recognition site is C-G-G motif in the loop structure that consists of seven nucleotides. But the putative IRE1 recognition sequence of miR-150 is not similar to this consensus. In addition, cleavage efficiency of miR-150 by IRE1 is considerably low as compared with that of XBP1 mRNA. The authors should show that stability of miR-150 is reduced and transcription of miR-150 does not change during fibrosis.

We thank the reviewer for bringing up this point. We have now added a stability assay to show that miR150 reduction is a consequence of IRE1 α cleavage and not a result of reduced transcription

(Figure 2M)^{19,20}. Some reports have also challenged the requirement for loop regions in the cleavage ability of IRE1 α , based mainly on sequence and localization²¹. We have identified several putative sites (TGCT) for cleavage by IRE1 α similarly to Upton *et al*²⁰. We believe that miR-150 cleavage by IRE1 α might prevent further processing by DICER, as suggested previously for IRE1 α ²⁰, or MCP1²², another ribonuclease. We have added this information to the manuscript.

<Typographical errors>

(1) Page 3, line 12: "in heart failure.." should be "in heart failure."

This was altered in the revised version manuscript.

Referee #3 (Comments on Novelty/Model System):

The manuscript included data from in vitro, in vivo and human sample experiments which are technically well done. They also use a combination of genetic and pharmacologic strategies, which is also a strength.

Referee #3 (Remarks):

In this paper the authors' report the role of Endoplasmic Stress in tissue fibrosis. Specifically they show that inhibiting IRE-1 alpha; pharmacologically, blocks TGF β ; activation of myofibroblasts and prevents liver and skin fibrosis in mouse models. They supplemented this data by using genetic strategies, using IRE-alpha; KO and IRE1a; mutants lacking endoribonuclease activity, to confirm the role IRE-1a in myofibroblast differentiation. Further they show data that suggests IRE-alpha; mediates its effects via its ribonuclease activity on miR 150 and through an XBP-1-dependent pathway. They conclude that targeting ER stress may be a viable therapeutic approach for tissue fibrosis.

General Comment: Fibrosis of tissues is a common problem for which there is incomplete understanding and limited treatment options. For these reasons the work reported in this paper is important. There has been increased interest in exploring the role of ER stress in tissue fibrosis and this paper provides further evidence for its importance. The paper is technically well done as it includes both genetic and pharmacologic data and has in vitro, in vivo and human data. The authors provide substantial evidence linking ER stress as a mediator of TGF β ; induced myofibroblast differentiation and tissue fibrosis. The evidence that IRE-alpha; activation is important for these effects is compelling and relatively novel. The data that IRE-alpha; mediates its effects thru miR-150 and ER expansion is less well-supported. While the authors show data that support these pathways, they lack key experiments which would prove IRE-1a is mediating its effects via miR-150 and c-MYB, or how ER volume is increasing. The use of two tissue fibrosis models and use of human fibroblast to establish biologic relevance is a strength of the paper. Since they used lung fibroblasts, use of a lung fibrosis model would strengthen the generalizability of the paper.

We would like to thank referee three for his careful evaluation of our manuscript and agree that the link between miR150, cMYB and myofibroblast activation was not addressed sufficiently in the original manuscript.

The transcription factor c-MYB represents the top target gene of miR-150 and several studies have shown that c-MYB expression is regulated by miR-150²³⁻²⁵. c-MYB is a known regulator of α SMA

expression²⁶⁻²⁸. To further establish the link between cMYB and α SMA in the revised manuscript, we transfected HFL-1 fibroblasts with siRNA targeting cMYB (Figure 2I). In line with previous studies²⁶⁻²⁸, silencing cMYB reduced TGF β -induced α SMA mRNA expression (Figure 2J).

We also agree that the use of a lung fibrosis model would have been a good addition to the manuscript and would strengthen the general applicability of our model. However due to the lack of ethical permission and time to perform these experiments, we are not able to fulfill this request. To support the general applicability of our model, we have used primary lung fibroblasts from patients with inflammatory lung diseases, such as cystic fibrosis (CF), chronic obstructive pulmonary disease (COPD) and asthma. These three diseases are characterized by a prolonged inflammatory response. Activated myofibroblasts play an established role in asthma and CF, causing the deposition of ECM and thereby contributing to tissue remodeling and disease progression. When we measured α SMA-expression in these primary lung fibroblasts, a significant increase was seen in those derived from asthma patients (Figure 6E). Interestingly, these cells also expressed higher levels of the UPR-marker BiP (Figure 6F) and treatment with 4 μ 8C seemed to decrease α SMA back to levels comparable to healthy controls (Figure 6E). This further supports the general applicability of our model. We did not see an effect on fibroblasts derived from CF and COPD patients, yet since these fibroblasts also did not show an increased expression of α SMA, it could be that these cells were not activated myofibroblasts.

Specific Comments:

1. In figure 2C, there is no difference in miR-150 in response to TGF β in the IRE-alpha; KO cells. This discrepancy in response compared to the experiments in 2A and 2B is not discussed and needs clarification.

This discrepancy stems from the fact that the results in Figure 2C were normalized to their respective vehicle. This has been clarified in the corresponding figure legend.

2. While it is clear from the data that inhibiting IRE1 α ; leads to increased mir150 expression, the authors have not provided conclusive data IRE1 α ; is mediating its fibrotic effects via miR150. For example does inhibiting mir150 in in IRE1 α -KO cells rescue α SMA expression?

We agree that this experiment would strengthen our hypothesis and transfected IRE1 α -/- MEF cells with an anti-miR150-5p miRNA inhibitor (Figure 2D). Silencing of miR150 increased TGF β -induced α SMA mRNA while in untransfected IRE1 α -/- MEF cells α SMA mRNA levels remained unaltered after TGF β treatment (Figure 2E). This suggests that silencing miR150 rescues the inhibitory effect on myofibroblast differentiation that was caused by the loss of function of IRE1 α .

3. There is no blot confirming overexpression of miR150 (Figure 2D).

We have added data on the mi-R150 expression levels after transfection in Figure 2F.

4. The blot showing inhibition of TGF β induced α SMA expression in miR150 overexpressing cells is not very convincing.

We agree with the reviewer's suggestion and have included a clearer gel demonstrating the TGF β -induced α SMA induction in miR150 overexpressing cells (Figure 2H).

5. There is no experiment that links mir150 inhibited α SMA expression directly to its effects on

cMYB. This is a key experiment in order to establish the mechanism by which mir150 is mediating its anti-fibrotic effects.

We agree that the link between miR150, cMYB and myofibroblast activation was not addressed sufficiently in the original manuscript and therefore we studied this for the revised version of the manuscript. To further establish the link between cMYB and α SMA in the revised manuscript, we transfected HFL-1 fibroblasts with siRNA targeting cMYB (Figure 2I). Silencing cMYB reduced TGF β -induced α SMA mRNA expression (Figure 2J).

References:

- [1] Wei Y, Wang D, Gentile CL, Pagliassotti MJ: Reduced endoplasmic reticulum luminal calcium links saturated fatty acid-mediated endoplasmic reticulum stress and cell death in liver cells. *Molecular and cellular biochemistry* 2009, 331:31-40.
- [2] Tamaki N, Hatano E, Taura K, Tada M, Kodama Y, Nitta T, Iwaisako K, Seo S, Nakajima A, Ikai I, Uemoto S: CHOP deficiency attenuates cholestasis-induced liver fibrosis by reduction of hepatocyte injury. *American journal of physiology Gastrointestinal and liver physiology* 2008, 294:G498-505.
- [3] Malhi H, Kaufman RJ: Endoplasmic reticulum stress in liver disease. *Journal of hepatology* 2011, 54:795-809.
- [4] Estornes Y, Aguilera MA, Dubuisson C, De Keyser J, Goossens V, Kersse K, Samali A, Vandenabeele P, Bertrand MJ: RIPK1 promotes death receptor-independent caspase-8-mediated apoptosis under unresolved ER stress conditions. *Cell death & disease* 2015, 6:e1798.
- [5] Lim MP, Devi LA, Rozenfeld R: Cannabidiol causes activated hepatic stellate cell death through a mechanism of endoplasmic reticulum stress-induced apoptosis. *Cell death & disease* 2011, 2:e170.
- [6] Desmouliere A, Redard M, Darby I, Gabbiani G: Apoptosis mediates the decrease in cellularity during the transition between granulation tissue and scar. *The American journal of pathology* 1995, 146:56-66.
- [7] Horowitz JC, Lee DY, Waghray M, Keshamouni VG, Thomas PE, Zhang H, Cui Z, Thannickal VJ: Activation of the pro-survival phosphatidylinositol 3-kinase/AKT pathway by transforming growth factor-beta1 in mesenchymal cells is mediated by p38 MAPK-dependent induction of an autocrine growth factor. *The Journal of biological chemistry* 2004, 279:1359-67.
- [8] Tanaka T, Yoshimi M, Maeyama T, Hagimoto N, Kuwano K, Hara N: Resistance to Fas-mediated apoptosis in human lung fibroblast. *The European respiratory journal* 2002, 20:359-68.
- [9] Tian L, He LS, Soni B, Shang HT: Myofibroblasts and their resistance to apoptosis: a possible mechanism of osteoradionecrosis. *Clinical, cosmetic and investigational dentistry* 2012, 4:21-7.
- [10] Stalder L, Heusermann W, Sokol L, Trojer D, Wirz J, Hean J, Fritzsche A, Aeschmann F, Pfanzagl V, Basselet P, Weiler J, Hintersteiner M, Morrissey DV, Meisner-Kober NC: The rough endoplasmic reticulum is a central nucleation site of siRNA-mediated RNA silencing. *The EMBO journal* 2013, 32:1115-27.
- [11] Daniels SM, Melendez-Pena CE, Scarborough RJ, Daher A, Christensen HS, El Far M, Purcell DF, Laine S, Gatignol A: Characterization of the TRBP domain required for dicer interaction and function in RNA interference. *BMC molecular biology* 2009, 10:38.
- [12] Li S, Liu L, Zhuang X, Yu Y, Liu X, Cui X, Ji L, Pan Z, Cao X, Mo B, Zhang F, Raikhel N, Jiang L, Chen X: MicroRNAs inhibit the translation of target mRNAs on the endoplasmic reticulum in Arabidopsis. *Cell* 2013, 153:562-74.
- [13] Wang XZ, Harding HP, Zhang Y, Jolicoeur EM, Kuroda M, Ron D: Cloning of mammalian Ire1 reveals diversity in the ER stress responses. *The EMBO journal* 1998, 17:5708-17.

- [14] Nuovo G, Lee EJ, Lawler S, Godlewski J, Schmittgen T: In situ detection of mature microRNAs by labeled extension on ultramer templates. *BioTechniques* 2009, 46:115-26.
- [15] de Candia P, Torri A, Gorletta T, Fedeli M, Bulgheroni E, Cheroni C, Marabita F, Crosti M, Moro M, Pariani E, Romano L, Esposito S, Mosca F, Rossetti G, Rossi RL, Geginat J, Casorati G, Dellabona P, Pagani M, Abrignani S: Intracellular modulation, extracellular disposal and serum increase of MiR-150 mark lymphocyte activation. *PloS one* 2013, 8:e75348.
- [16] Yeh YY, Ozer HG, Lehman AM, Maddocks K, Yu L, Johnson AJ, Byrd JC: Characterization of CLL exosomes reveals a distinct microRNA signature and enhanced secretion by activation of BCR signaling. *Blood* 2015, 125:3297-305.
- [17] Zhang Y, Liu D, Chen X, Li J, Li L, Bian Z, Sun F, Lu J, Yin Y, Cai X, Sun Q, Wang K, Ba Y, Wang Q, Wang D, Yang J, Liu P, Xu T, Yan Q, Zhang J, Zen K, Zhang CY: Secreted monocytic miR-150 enhances targeted endothelial cell migration. *Molecular cell* 2010, 39:133-44.
- [18] Arroyo JD, Chevillet JR, Kroh EM, Ruf IK, Pritchard CC, Gibson DF, Mitchell PS, Bennett CF, Pogosova-Agadjanyan EL, Stirewalt DL, Tait JF, Tewari M: Argonaute2 complexes carry a population of circulating microRNAs independent of vesicles in human plasma. *Proceedings of the National Academy of Sciences of the United States of America* 2011, 108:5003-8.
- [19] Lerner AG, Upton JP, Praveen PV, Ghosh R, Nakagawa Y, Igarria A, Shen S, Nguyen V, Backes BJ, Heiman M, Heintz N, Greengard P, Hui S, Tang Q, Trusina A, Oakes SA, Papa FR: IRE1alpha induces thioredoxin-interacting protein to activate the NLRP3 inflammasome and promote programmed cell death under irremediable ER stress. *Cell metabolism* 2012, 16:250-64.
- [20] Upton JP, Wang L, Han D, Wang ES, Huskey NE, Lim L, Truitt M, McManus MT, Ruggero D, Goga A, Papa FR, Oakes SA: IRE1alpha cleaves select microRNAs during ER stress to derepress translation of proapoptotic Caspase-2. *Science* 2012, 338:818-22.
- [21] Gaddam D, Stevens N, Hollien J: Comparison of mRNA localization and regulation during endoplasmic reticulum stress in *Drosophila* cells. *Molecular biology of the cell* 2013, 24:14-20.
- [22] Suzuki HI, Arase M, Matsuyama H, Choi YL, Ueno T, Mano H, Sugimoto K, Miyazono K: MCP1 ribonuclease antagonizes dicer and terminates microRNA biogenesis through precursor microRNA degradation. *Molecular cell* 2011, 44:424-36.
- [23] Feng J, Yang Y, Zhang P, Wang F, Ma Y, Qin H, Wang Y: miR-150 functions as a tumour suppressor in human colorectal cancer by targeting c-Myb. *Journal of cellular and molecular medicine* 2014, 18:2125-34.
- [24] Lin YC, Kuo MW, Yu J, Kuo HH, Lin RJ, Lo WL, Yu AL: c-Myb is an evolutionary conserved miR-150 target and miR-150/c-Myb interaction is important for embryonic development. *Molecular biology and evolution* 2008, 25:2189-98.
- [25] Yang K, He M, Cai Z, Ni C, Deng J, Ta N, Xu J, Zheng J: A Decrease in miR-150 Regulates the Malignancy of Pancreatic Cancer by Targeting c-Myb and MUC4. *Pancreas* 2015, 44:370-9.
- [26] Kitada T, Seki S, Nakatani K, Kawada N, Kuroki T, Monna T: Hepatic expression of c-Myb in chronic human liver disease. *Hepatology* 1997, 26:1506-12.
- [27] Buck M, Kim DJ, Houglum K, Hassanein T, Chojkier M: c-Myb modulates transcription of the alpha-smooth muscle actin gene in activated hepatic stellate cells. *American journal of physiology Gastrointestinal and liver physiology* 2000, 278:G321-8.
- [28] Zheng J, Lin Z, Dong P, Lu Z, Gao S, Chen X, Wu C, Yu F: Activation of hepatic stellate cells is suppressed by microRNA-150. *International journal of molecular medicine* 2013, 32:17-24.

We have now received the enclosed reports from the referees that were asked to re-assess it. As you will see the reviewers are now globally supportive and I am pleased to inform you that we will be able to accept your manuscript pending the following final amendments:

1) We are now encouraging the publication of source data, particularly for electrophoretic gels and blots, with the aim of making primary data more accessible and transparent to the reader. Would you be willing to provide a PDF file per figure that contains the original, uncropped and unprocessed scans of all or at least the key gels used in the manuscript? The PDF files should be labeled with the appropriate figure/panel number, and should have molecular weight markers; further annotation may be useful but is not essential. The PDF files will be published online with the article as supplementary "Source Data" files. If you have any questions regarding this just contact me.

2) Every published paper now includes a 'Synopsis' to further enhance discoverability. Synopses are displayed on the journal webpage and are freely accessible to all readers. They include a short standfirst as well as 2-5 one sentence bullet points that summarise the paper. Please provide the synopsis including the short list of bullet points that summarise the key NEW findings. The bullet points should be designed to be complementary to the abstract - i.e. not repeat the same text. We encourage inclusion of key acronyms and quantitative information. Please use the passive voice. Please attach this information in a separate file or send them by email, we will incorporate it accordingly. You are also welcome to suggest a striking image or visual abstract to illustrate your article. If you do please provide a jpeg file 550 px-wide x 400-px high.

3) I note that Fig.s 1D and 5K are excessively contrasted. Please modify by reducing the contrasting and provide the source data for these images (in addition to the others should you wish to do so).

Please submit your revised manuscript within two weeks. I look forward to seeing a revised form of your manuscript as soon as possible.

***** Reviewer's comments *****

Referee #2 (Remarks):

The authors fully improved the manuscript, and the reviewer thinks that it is now suitable for publication in the journal of EMBO Molecular Medicine.

Referee #3 (Comments on Novelty/Model System):

The experiments are logical, well done and include in vitro, animal and human data.

Referee #3 (Remarks):

In this revised paper the author's report the relationship between ER stress, IRE1-a mediated degradation of miR-150, XBP-1 splicing and tissue fibrosis. The authors' address my previous concerns about showing relevance to lung diseases and most of my other specific comments.

Corresponding Author Name: Pär Gerwins

Journal Submitted to: EMBO Molecular medicine

Manuscript Number: EMM-2015-05925